# Triterpene and Steroid Glycosides from Marine Sponges (Porifera, Demospongiae): Structures, Taxonomical Distribution, Biological Activities

**DOI:** 10.3390/molecules28062503

**Published:** 2023-03-09

**Authors:** Natalia V. Ivanchina, Vladimir I. Kalinin

**Affiliations:** G.B. Elyakov Pacific Institute of Bioorganic Chemistry, Far-Eastern Branch of the Russian Academy of Sciences, Prospect 100 Letya Vladivostoka, 159, 690022 Vladivostok, Russia

**Keywords:** tetracyclic triterpene glycosides, steroid glycosides, taxonomical distribution, biological role, biological activities, marine sponges, Demospongiae

## Abstract

The article is a comprehensive review concerning tetracyclic triterpene and steroid glycosides from sponges (Porifera, Demospongiae). The extensive oxidative transformations of the aglycone and the use of various monosaccharide residues, with up to six possible, are responsible for the significant structural diversity observed in sponge saponins. The saponins are specific for different genera and species but their taxonomic distribution seems to be mosaic in different orders of Demospongiae. Many of the glycosides are membranolytics and possess cytotoxic activity that may be a cause of their anti-predatory activities. All these data reveal the independent origin and parallel evolution of the glycosides in different taxa of the sponges. The information concerning chemical structures, biological activities, biological role, and taxonomic distribution of the sponge glycosides is discussed.

## 1. Introduction

Sponges (phylum Porifera) are an oldest living group of Metazoa. Their general organization was so adaptable that they survived during dramatic changes in environment. This group of aquatic organisms (marine and freshwater) are very diverse and includes more than 8500 valid species [1,2]. The unique conditions of the marine environment, such as high pressure and salinity, coupled with the need to defend against predators and microorganisms, have led to a wide range of secondary metabolites in sponges. These metabolites possess medicinal properties that are distinct from those found in terrestrial plants [3]. The number of such metabolites found in sponges by now is more than 5300. Some of them are anticancer agents which have different mechanisms of action including anti-neoplastic efficacy [4,5]. Furthermore, the metabolites obtained from marine sponges may have extensive medicinal applications, including preparations with antiviral and anti-inflammatory properties, as referenced in [6] and [7], respectively.

It is known that many sponge metabolites may be synthesized by symbiotic microorganisms [3]. However, the enzymes responsible for terpene biosynthesis were found recently in sponges but not in symbiotic microorganisms. Hence, these metabolites should be synthesized by sponges directly [8]. The distribution of different groups of secondary metabolites within the class Demospongiae of the phylum Porifera (sponges) was used for chemotaxonomy purposes. The authors of the investigation suggest that the distribution reveals polyphyly of any taxonomical groups and the system of the class should be improved [9].

Tetracyclic steroid and triterpene glycosides are characteristic substances for many terrestrial higher plants. Over the past few decades, a large number of triterpene glycosides have been identified in animals, specifically sea cucumbers (Holothurioidea, Echinodermata) from various orders, exhibiting taxonomic specificity [10,11,12,13,14,15]. Another well-known group of bioactive marine glycosides is steroid glycosides from starfish (Asteroidea, Echinodermata) [16,17,18,19]. Terpenoid and steroid glycosides in marine sponges were discovered during the 1980s and early 1990s. The last review partially covered sponge triterpene and steroid glycosides has been published in 2012 [20]. In this review, we discuss the data from the literature which concerns the main trends in the investigations on sponge tetracyclic triterpene and steroid glycosides, including chemical structures, biological activities, and taxonomic distribution, along with the last taxonomical revisions.

## 2. Tetracyclic Triterpene Glycosides

### 2.1. The Order Tetractinellida (Suborder Astrophorina)

The sponge *Melophlus sarasinorum* (family Geodiidae, subfamily Erylinae) is very common in the Indo-West Pacific tropical region. This sponge also has such junior synonymic names as *Asteropus sarasinorum*, *Melophlus isis* and *Stellettinopsis isis* [1]. Moreover, there are erroneous names for this using in chemical articles including *Asteropus sarasinosum* [21,22] and *Melophlus sarasinorum* [23]; however, we apply only the valid taxonomical name in this article (see Section 4. Taxonomic Distribution of Glycosides in Sponges, Table 1).

The glycosides of *Melophlus sarasinorum* form a mixture of 14-nor-methyl-lanostane (or 30-norlanostane) derivatives. Nine glycosides, sarasinosides A_1_–A_3_ (**1**–**3**), B_1_–B_3_ (**4**–**6**), and C_1_–C_3_ (**7**–**9**), were firstly found by Kitagawa et al. [21,24] from specimens collected in shallow waters of the Palauan Archipelago. Sarasinoside A_1_ (**1**) was also found by Schmitz et al. [22] from the sponges harvested near Guam Island and Truk Lagoon. Other similar glycosides, sarasinosides D–G (**10**–**13**), as well as known sarasinoside B_1_ (**4**), were found by Espada et al. in the same species harvested near Guam Island [25] (Figure 1).

Four new 14-nor-methyl-lanostane glycosides, sarasinosides H_1_ (**14**), H_2_ (**15**), I_1_ (**16**), and I_2_ (**17**), were isolated by Lee et al. These authors also have isolated two known sarasinosides A_1_ (**1**) and A_3_ (**3**) from the same species, harvested in Guam shallow waters [26]. Dai et al. have obtained four new glycosides, sarasinosides J (**18**), K (**19**), L (**20**), and M (**21**), together with sarasinosides A_1_ (**1**), A_3_ (**3**), H_2_ (**15**), I_1_ (**16**), and I_2_ (**17**), from the specimens harvested near Sulawesi, Indonesia [23]. Santalova et al. [27] isolated two similar triterpene glycosides, sarasinosides A_4_ (**22**) and A_5_ (**23**). They have also isolated four known glycosides, sarasinosides A_1_–A_3_ (**1**–**3**), L (**20**), and M (**21**), from the same Australian collection of *M. sarasinorum* (Figure 1).

Sarasinoside M_2_ (**24**), which is similar to the sarasinoside glycoside, along with earlier known sarasinoside B_1_ (**4**), was isolated from an unidentified sponge harvested near the Solomon Islands. Sarasinoside M_2_ possesses the same aglycone as sarasinoside M (**21**) although the third (preterminal) glucose residue in low semi-chain of its sugar moiety is replaced by xylose [28] (Figure 1).

The most aglycones of studied sarasinosides possess a norlanostane polycyclic system having 8(9)-, 9(11)- or 8(14)-double bonds and identical 23-keto-Δ^24(25)^ side chains, whereas glycosides **2**, **5,** and **8** possess 7(8),9(11)-diene system, but the substances **3**, **6**, and **9** have 8(9),14-diene system. The most uncommon aglycone of sarasinoside D (**10**) contains an additional hydroxyl attached to C-12, a saturated core, and a methyl group attached to C-8 instead of common C-14 position. A similar arrangement of methyl group, though Δ^14(15)^-unsaturation, was found in some triterpenoids of higher plants [25]. More oxidized aglycones in sarasinosides E, F, H_1_, H_2_, I_1_, I_2_, J–L, and A_5_ (**11**, **12**, **14**–**20**, **23**, respectively) contain additional hydroxy-, methoxy- or keto-groups, whereas the aglycones of sarasinosides M, M_2_, and A_4_ (**21**, **24**, **22**) are 8,9-seco-derivatives that possess very uncommon 8α,9α-epoxy-8(14),9(11),24-triene and 8α,9α-epoxy-7(8),9(11),24-triene (Figure 1) [23,24,25,26,27,28].

Carbohydrate chains of all the glycosides have the same architecture. All monosaccharide residues (xylose, glucose, N-acetyl-2-deoxy-2-amino-galactose, as well as N-acetyl-2-deoxy-2-amino-glucose) belong to D-series and are in pyranose forms. The glycosidic centers have β-configuration. Glycosides **1**–**3**, **14**–**23** are pentaosides with glucose residue as the third monosaccharide, glycosides **4**–**6**, **10**–**13**, as well as **24** having xylose instead of glucose. Nevertheless, glycosides **7**–**9** are tetraosides with identical carbohydrate chains having xylose as the third monosaccharide residue. This monosaccharide occupies terminal position.

In total, **23** sarasinosides were isolated from *Melophlus sarasinorum*, and one of their congeners was also found in an unidentified sponge. Most of them are pentaoside glycosides and their carbohydrate moieties may differ in the third monosaccharide residue (xylose or glucose) only. The cause of structural diversity of sarasinosides seems to be both oxidative processes in rings B, C, and D and an occupation of different positions by double bonds in the tetracyclic systems of aglycones.

Sarasinosides possess ichthyotoxic and cytotoxic properties. Kitagawa et al. investigated ichthyotoxic activities of sarasinosides A_1_ (**1**) and B_1_ (**4**) on killifish *Poecilia reticulata* [21]. Sarasinosides A_1_ (**1**), having glucose as the third sugar, revealed LD_50_ of 0.39 μg/mL, but glycoside sarasinoside B_1_ (**4**), having xylose residue in the same position, revealed LD_50_ of 0.71 μg/mL. These glycosides have moderate inhibitory activities (ED_50_ of 10 μg/mL) on the fertilized eggs of the starfish *Asterina (=Patiria) pectinifera*. Schmitz et al. have found the cytotoxicity of glycoside **1** on murine lymphocytic leukemia P388 cell line (ED_50_ of 2.8 μg/mL) [22]. Lee et al. have reported cytotoxic activities of glycosides **2** and **3** on human leukemia cell line K562 (ED_50_ of 6.5 and 17.1 μg/mL, respectively), whereas glycosides **14**–**17** were not active [26]. Dai et al. reported that sarasinoside A1 (**1**) showed potent activity against the yeast *Saccharomyces cerevisiae*, but had no effect on the bacteria *Escherichia coli* and *Bacillus subtilis*. Sarasinoside J (**18**) was also very active on *S. cerevisiae*. However, it has had a moderate activity on *B. subtilis* [23]. Sarasinoside M_2_ (**24**) revealed moderate cytotoxicity toward Neuro-2a and HepG2 tumor cell lines (EC_50_ of 17 and 19 μM respectively) [28]. Hence, sarasinosides with common 8(9)-unsaturation or 7(8),9(11)-diene systems have strong or moderate cytotoxic effects on yeast, fertilized eggs of starfish, and tumor cell lines, and furthermore are ichthyotoxic. Biosynthetic transformations of C and D rings, including the migration of a double bond into the 8(14)-position, followed by the introduction of oxygen-bearing substituents, along with other oxidative reactions in aglycone cyclic systems, decrease the activities.

The study of marine sponges of the genus *Erylus* (family Geodiidae, subfamily Erylinae) began in the late 1980s of the last century and continues to the present time. These studies began with the study of the sponge *Erylus lendenfeldi*, which is a habitant of the Indo-West Pacific zoogeographical region. Eryloside A (**25**), a new bis-nor-triterpene bioside, with 4β,14-di-nor-methyl-lanostane aglycone and β-D-galactopyranose as the first sugar together with another (terminal) β-D-galactopyranose linked to C-2 of the first sugar, was found in the Red Sea sample of this sponge species by Carmely et al. in 1989 [29]. Glycoside **25** aglycone possesses 8(9)- and 14(15)-double bonds and a hydroxyl at C-23.

Later, Fouad et al. [30] isolated two related glycosides named as erylosides K and L (**26**, **27**), in addition to **25** from the same sponge, which was harvested near Jordan’s coast in the Gulf of Aqaba, Red Sea. The absolute stereochemistry of C-23 in eryloside K was established by comparison the NMR data with those published for both possible diastereomers, which exact stereochemistry has been established by Mosher’s method. Eryloside L (**27**) had very uncommon 8α,9α-epoxy-8,9-seco-7,9(11),14-triene fragment in the aglycone (Figure 2) [30].

Almost simultaneously, Sandler et al. [31] from Faulkner’s laboratory have isolated the same eryloside A (**25**) with two similar glycosides named erylosides K and L (**26**, **28**) from another Red Sea collection of this species. The structure of eryloside K was identical to that described by Fouad et al. [30]. The second new glycoside **28**, also called eryloside L by Sandler et al. [31], differed in structure from the compound described by Fouad et al. [30]. Eryloside L1 (**28**), depicted in Figure 2, contains an aglycone similar to that of erylosides A and K, as well as a 23-keto group in the side chain. The 23*S*-configuration of eryloside A (**25**) was firstly established by Mosher’s method. The C-23 absolute configuration in **26** was assigned by comparison ^1^H NMR spectra, HPLC retention times, and optical rotation derivatives of **25** and **26** obtained by hydrogenation using a rhodium catalyst [31].

Glycoside **25** reveals antifungal action on *Candida albicans* (MIC of 15.6 μg/mL) and cytotoxicity on P388 tumor cell line (IC_50_ of 4.2 μg/mL) [30]. This glycoside is active on *Bacillus subtilis, C. albicans*, and *Escherichia coli* (zones of inhibition at 10 μg per disc were of 7, 7, and 6 mm, respectively) [30]. Glycosides **25** and **26** gave the mortality rate of 50% at the concentration of 0.1 μg/mL in the brine shrimp assay. Glycoside **27** was inactive in this test. Glycosides **25**, **26**, and **27** are not cytotoxic on THP-1, JURKAT, and MM-1 tumor cell lines [30]. Glycosides **25**, **26**, and **28** were selectively active on the Δrad50 budding yeast strain (0.8, 2.0, and 3.4 μg/mL) in comparison with the action on the wild parent yeast strain (IC_50_ of 3.5, 6.1, and 11.4 μg/mL, respectively). The selective cytotoxicity against similar mutant yeast is a probable indicator of topoisomerase inhibitors. However, the activities on TOP1oe (IC_50_ of 5.7, 7.5, and 10.9 μg/mL) and TOP2oe (10.8, 12.2, and 9.5 μg/mL) were weaker than for camptothecin and idarubicin, which are known inhibitors. The aglycone, obtained from glycoside **25**, was inactive against the both yeast strains. This finding confirmed the significant role of the carbohydrate chain in the activity, as reported in reference [31].

The studies on a sponge *Erylus* sp. collected at a depth of 500 m from New Caledonian waters yielded two new lanostane glycosides: trioside and tetraoside and erylosides C (**29**) and D (**30**), respectively [32]. Their aglycones have 8(9)-unsaturation, a carboxyl at C-14, a methylene at C-24 and an additional methyl at C-25. All the sugars are D-galactose residues in β-pyranose form. One terminal monosaccharide residue is linked to C-2 of the first monosaccharide. Another terminal sugar is attached to C-3 of the first monosaccharide residue. This is characteristic for many triterpene glycosides isolated from sponges belonging to the genus *Erylus*. Nevertheless, one of the terminal monosaccharide residues in tetraoside **30** is linked to C-4 of the second monosaccharide residue, but not to C-3 position (Figure 2).

The sponge *Erylus goffrilleri* is an inhabitant of the tropical shallow waters of the Atlantic. Eryloside E (**31**), an unusual lanostane glycoside, was isolated by Gulavita et al. It contains a bioside carbohydrate chain consisting of galactose as the first monosaccharide, with N-acetyl-2-deoxy-2-amino-glucose linked to C-2 of the galactose residue. It also has another carbohydrate moiety composed of xylose, attached to a carboxyl at C-14 of the aglycone [33]. The aglycone possesses 8(9)-double unsaturation, oxy-group at C-24, and additional methyls at C-25, as well as C-24 (Figure 3).

Afiyatullov et al. discovered four lanostane monosides, erylosides R (**32**), S (**33**), T (**34**), and U (**35**), which are similar glycosides with a β-D-galactopyranose residue serving as a carbohydrate moiety, as depicted in Figure 3 [34]. Glycoside **32** contains a lanostane derivative as an aglycone. It has 8(9)-double unsaturation, carboxyl at C-14, hydroxyl at C-24 and two additional methyls at C-25, as well as C-24 in the side chain. Eryloside S (**33**) contains a relative aglycone, having an acetate and an additional methyl group at C-24 in the side chain. The aglycone of eryloside T (**34**) has the same side chain as in **32** but possesses 7(8)-unsaturation and an uncommon lactone linkage between C-14 and C-9. Glycoside **35** is similar to **34** but has an additional 7,8-epoxy group. The configuration of this epoxy-group was indicated by the authors of the present paper as 7α,8α on the basis of the upfield shift of the C-5 signal [34]. Later, Kolesnikova et al. isolated a series of free similar aglycones from a sponge *Penares* sp. which was collected by dredging in Vietnamese waters. They determined the stereochemistry of these substances using X-ray analysis and CD spectra and found that the aglycones investigated contained a 7β,8β-epoxy group [55]. As a result, the structure of **35** was revised and presented at Figure 3 with a 7β,8β-epoxy group.

Moreover, Afiyatullov et al. found three lanostane biosides: erylosides F_5_–F_7_ (**36**–**38**), as well as eryloside V (**39**), and a trioside (Figure 3). The carbohydrate chains of glycosides **36** and **37** are very similar, consisting of D-galactose and N-acetyl-2-deoxy-2-amino-D-glucose, and are linked to C-2 of the galactose [34]. The sugars are in β-pyranose forms, and their aglycones have an 8(9)-double bond and a carboxy-group at C-14, which is common for many erylosides. As it also is characteristic for many erylosides, the side chain of the aglycone in glycoside **36** contains a C-24 hydroxyl and two additional methyls at C-25, as well as at C-24. The side chain of **37** contains a 24-OAc-group and only one additional C-24 methyl group. Glycoside **38** has the same aglycone as in **36**. However, it has the second monosaccharide residue, β-D-glucopyranose, linked to C-3 of the first sugar. Eryloside V (**39**) is a trioside having the same aglycon as in **36** and α-L-arabinopyranose as the first monosaccharide residue and β-D-galactopyranose and β-D-xylopyranose at C-2 and C-3 of arabinose residue, respectively. A specific structural feature of erylosides from this sponge is additional alkylation in the side chain.

Glycoside **31** weakly inhibited the binding of ^125^[I]-Bottom Hunter labeled C5a to its receptor (IC_50_ > 10 μM). Eryloside E (**31**) reveals also immunosuppressive activity (EC_50_ of 1.3 μg/mL). Its immunosuppressive action was specific and independent from general cytotoxicity [33]. Erylosides R–T (**32**–**34**), F_6_ (**37**), F_7_ (**38**), and V (**39**) were cytotoxic against Ehrlich carcinoma tumor cells (IC_50_ of 20–40 μM) [34]. Nevertheless, glycosides **35** and **36** were not active.

In continuation of the investigation of *E. goffrillery*, which was harvested from the Caribbean Sea near the Arrecife-Seco reef (Cuba), Antonov et al. [35] isolated seven new tetracyclic triterpene glycosides: erylosides F_8_ (**40**), V_1_ (**41**), V_2_ (**42**), V_3_ (**43**), W (**44**), W_1_ (**45**), and W_2_ (**46**) (Figure 3). Erylosides **40** and **43** have 14-carboxy-24,25-dimethyllanost-8(9)-en-3β,25-diol as aglycone whereas glycosides **41**, **42**, and **44**–**46** possess 14-carboxy-24-acetoxy-24-methyllanost-8(9)-en-3β-ol as aglycone. Glycoside **40** is a bioside having β-D-galactopyranose as a first monosaccharide residue attached to C-3 of the aglycone and the same monosaccharide residue attached to C-2 of the first galactopyranose. Glycosides 41, 42, and 43 are triosides consisting of the same first sugar, with the second and third monosaccharides attached to C-2 and C-3 of the first sugar, respectively. Glycoside **41** has β-D-galactopyranose as the second sugar whereas **42** and **43** have N-acetyl-2-deoxy-2-amino-β-D-galactopyranose. Glycoside **41** has xylose as the third monosaccharide residue whereas **42** and **43** have α-L-arabinopyranose. Glycosides **44**–**46** are tetraosides. The first sugar of **44** is β-D-galactopyranose whereas **45** and **46** have 6-OAc-β-D-galactopyranose. Two residues of α-L-arabinopyranose, attached to each other by 1→4 glycosidic link, are linked to C-3 of the first sugar for all the tetraosides **44**–**46**. The fourth (terminal) sugar for **44** and **45** attached to C-2 of the first monosaccharide residue is N-acetyl-2-deoxy-2-amino-β-D-galactopyranose whereas N-acetyl-2-deoxy-2-amino-β-D-glucopyranose is the fourth sugar for glycoside **46** [35].

Many glycosides of this series have moderate cytotoxicity and hemolytic activities. The activities depend on both aglycone and carbohydrate moieties [35]. Erylosides V_2_ (**42**) and V_3_ (**43**) have the same trioside carbohydrate moieties but different aglycones. They reveal quite different activities. Eryloside V_2_ (**42**), having 24-*O*-acetyl and having no methyl at C-25, is a moderate hemolytic. Nevertheless, it possesses significant cytotoxic activities. Glycoside **43,** having a 24-hydroxyl and a C-25 methyl, has both moderate hemolytic and cytotoxic activities. Eryloside V_2_ (**42**), which is a trioside, is more active in both tests than eryloside W (**44**), which is a tetraoside that has an additional L-arabinose. The activities of glycoside **44,** having D-galactose as the first sugar, are significantly lower than the activities of **45**, which possesses a 6-*O*-acetyl group of the first D-galactose. The activity of glycoside **45** with N-acetyl-D-galactosamine as the fourth sugar are higher than the activities of **46** with an N-acetyl-D-glucosamine at the same position. It is interesting that glycoside **42** possesses strong cytotoxic and moderate hemolytic activities. However, glycoside **43** has moderate cytotoxic activity and low hemolytic activity [35]. This indicates a more complicated character of cytotoxicity for these glycosides than a simple membranolytic action.

The aglycone present in almost all glycosides isolated from *E. goffrilleri* is a lanostane derivative with a double bond at position 8(9) and a carboxyl group at C-14. This tetracyclic triterpene nucleus was found earlier from *Penares* sp. as penasterol, a free non-glycosylated triterpene [56]. 3-oxo- and 3-*O*-acetyl-derivatives of penasterol were also found in the sponge *Penares incrustans* [57]. Only two types of side chains were found in them: the first one contains a C-24 hydroxyl, along with two additional methyls at C-25 and C-24; the second one contains an acetyl group and one additional C-24 methyl.

All erylosides from *E. goffrilleri* possess a similar architecture of the carbohydrate chain, and all the sugars are in pyranose form. The first monosaccharide residue attached to C-3 of a 14-carboxylated aglycone is β-D-galactose. Sometimes it may be a 6-*O*-acetyl-β-D-galactose or α-L-arabinose. The sugar number is varied from one to four. Next, monosaccharide residues may be linked to the first sugar C-2 and C-3 positions. Further elongation of the carbohydrate chain may occur only from the sugar that is attached to C-3 of the first sugar. The next (terminal) monosaccharide residue may be attached only to C-4 of this second sugar. A terminal monosaccharide residue attached to the first sugar C-2 position may be D-galactose, N-acetyl-D-galactosamine, or N-acetyl-D-glucosamine. The configurations of the glycosidic centers are α for L-arabinose and β for D-glucose, D-galactose, 6-*O*-acetyl-D-galactose, N-acetyl-D-glucosamine, N-acetyl-D-galactosamine, as well as D-xylose.

The sponge *Erylus formosus* is a typical habitant of shallow waters from the Caribbean Sea and eastern Brazil. It is the most investigated sponge concerning tetracyclic triterpene glycosides. In continuation of investigations into this species, Jaspars and Crews have found formoside (**47**), a lanostane tetraoside. It was isolated from the sample of this sponge harvested in shallow waters of Bahamas [36] (Figure 4). The aglycone of formoside is a lanostane derivative, with 8(9)- and 24(25)-double bonds and C-14 carboxyl. Such structural features are characteristic for penasterol [56]. All monosaccharide residues (two galactoses and two arabinoses) of **47** are in pyranose forms.

Another glycoside of this type, formoside B (**48**), was isolated by Kubanek et al. in the extract of *E. formosus*, also collected near the Bahamas, along with known formoside (**47**) (Figure 4) [37]. Glycoside **48** differs from **47** because of the presence of terminal N-acetyl-2-deoxy-2-amino-D-galactose instead of a galactose residue. The authors also characterized, using mass-spectrometry without the isolation of individual glycosides, a series of inseparable hexaoside and triosides fractions [37].

Stead et al. discovered eryloside F (**49**), a similar lanostane bioside, as a result of their investigation into *E. formosus*. The sponge was harvested near the Bahamas through scuba diving at a depth of 55 feet. It has the same aglycone, also containing arabinose residue as the first monosaccharide residue linked to C-3 of the aglycone and the second sugar (galactose) attached at the arabinose C-2 position [38] (Figure 4).

The extremely hard task of separation of very complicated mixtures was successfully resolved by Antonov et al. They have isolated a series of individual lanostane biosides, triosides, and hexaosides [39,40] from collection of this sponge from the Mexican Gulf (Figure 4). All the sugars in these glycosides are in β-pyranose forms, belong to D-series, except arabinose residues, which belong to L-series. The last ones have an α-configuration of glycosidic bonds. Erylosides F_1_–F_4_ (**50**–**53**) were similar to eryloside F (**49**) and differing by aglycone side chains [39]. The known eryloside F (**49**) also was found. Eryloside F_1_ (**50**) possesses a penasterol such as aglycone having the side chain with a C-24 methylene. Penasterol and the discussed relative aglycones are predominant in the *Erylus* spp. glycosides. Eryloside F_2_ (**51**) possesses a 25(26)-double bond and a 24R-hydroxy-group. Eryloside F_3_ (**52**) has a 25(26)-double bond and 24S-hydroxyl. Eryloside F_4_ (**53**) is a 24-keto-derivative of erylosides F_2_ (**51**) and F_3_ (**52**) [39].

Erylosides M (**54**) and N (**55**), two new triosides, were also found by the present authors in *E. formosus* [39], as well as eryloside H (the structure is shown below in Figure 5), earlier found in *Erylus nobilis* [41]. Eryloside N (**55**) has penasterol as an aglycone. However, eryloside M (**54**) contains an aglycone with a C-24 methylene group in the side chain. Eryloside O (**56**), a new tetraoside is similar to formoside (**47**), but the third terminal sugar was identified as α-L-arabinose but not β-D-galactose. Erylosides P (**57**) and Q (**58**), two new hexaosides, have the same carbohydrate chains. They differed by the aglycones side chains. The carbohydrate chains of these glycosides are very closed to that of formoside (**47**). However, they have an additional glucose residue linked to C-4 of the fourth monosaccharide residue (galactose) and a terminal xylose, attached to this glucose C-2 position [39].

Antonov et al. continued the investigations and discovered eryloside R_1_ (**59**), a trioside, along with formoside (**47**) and six new hexaosides, erylosides T1–T6 (**60**–**65**) [40] (Figure 4). The sugars in all these glycosides were in β-D-pyranose forms except for L-arabinose residues, which had an α-configuration. Eryloside R_1_ (**59**) has penasterol as its aglycone. Its terminal galactose is attached to C-3 of the first sugar (arabinose), whereas another galactose residue is linked to the arabinose C-2 position. Hexaosides **60**–**62** also contain penasterol as the aglycone. However, hexaosides **63**–**65** contain an aglycone with a methylene group at C-24 in the side chains which is common in this series [40]. All the hexaosides possess carbohydrate moieties as they have the same architecture (general structural plan). Their carbohydrate chains are closed to those of erylosides P (**57**) and Q (**58**) [39].

It is interesting that all the triterpene glycosides found in *E. formosus* (totally 20 glycosides) have similar structural features. All the glycosides contain α-L-arabinopyranose as the first sugar linked to an aglycone C-3 position. The number of sugars may vary from two to six.

One of terminal monosaccharides is attached at C-2 of the first sugar (L-arabinose). All the hexaosides also contain another terminal sugar, linked to C-2 of the fourth monosaccharide residue. All other links between the sugars are 1 → 3, except a link between the third and fourth as well as the fourth and fifth sugars, which is 4 → 1 and 2 → 1, respectively. Figure 4 shows that N-acetyl-2-deoxy-2-amino-D-galactose may also serve as a terminal monosaccharide in tetracyclic triterpene glycosides of E. formosus, in addition to arabinose, glucose, and galactose.

There is a possibility that the biosynthesis of carbohydrate chains in E. formosus is initiated by a glycosyltransferase containing L-arabinose. Through the use of other glycosyltransferases, elongation of the carbohydrate chains is possible, which allows for the substitution of different sugars in various triterpene glycosides while maintaining the overall structure of the carbohydrate chains [40]. Eryloside F (**49**) is a potent thrombin receptor antagonist. It may inhibit human platelet aggregation in vitro. Glycoside **49** facilitates the Ca^2+^ mobilization in cells registered by an FLIPR assay [37]. Formoside (**47**) revealed antiviral effect (IC_50_ of 3.5 μg/mL vs. HSV-1) and moderate antibacterial activity (IC_50_ of 31.3 μg/mL vs. *Corynebacterium xerosis*) [36]. Hexaoside fractions have an antifungal effect on *Candida albicans* resistant to amphotericin B (IC_50_ of 3.9 μg/mL) [38]. At comparable concentrations, eryloside F1 (**50**) and eryloside F (**49**) both elicit a stimulation of calcium influx into mouse splenocytes (130% of the control), whereas eryloside F3 (**52**) induces early apoptosis in Ehrlich cells at a concentration of 100 μg/mL, whereas its epimer eryloside F2 (**51**) does not have this effect [39].

Erylosides G-J (**66**–**69**), which are lanostane triosides, were obtained by Shin et al. from the sponge Erylus nobilis. The sponge was collected from shallow waters surrounding Jaeju Island in Korea [41]. The lanostane aglycones possess an 8(9)-double bond, C-14 carboxyl, and 24-methylene in the side chains (Figure 5). Glycosides **68** and **69** also possess an additional methyl at C-25 position, similarly to the glycosides from *E. goffrilleri*. Triterpene glycosides **66**–**69** are triosides that exhibit moderate cytotoxic activity against the human leukemia cell line K562, with IC50 values of 22.1, 24.8, 17.9, and 21.8 μg/mL, respectively [41].

Japanese authors have found nobiloside **70**, a linear lanostane trioside, from *E. nobilis* harvested in shallow waters of Shikine-jima Island, Japan [42]. Nobiloside aglycone has 8(9)- and 24(25)-double bonds and C-14 carboxyl (Figure 5). The first monosaccharide residue in the carbohydrate moiety is β-D-glucuronic acid in pyranose form, the second one is β-D-galacturonic acid, and α-L-arabinose is a terminal sugar. Nobiloside (**70**) reveals an inhibition of the bacterium *Clostridium perfringens* neuraminidase (IC_50_ of 0.46 μg/mL).

A study [58] has reported the synthesis of two trisaccharides containing D-galactose, L-arabinose, and D-glucosamine hydrochloride, which are structurally similar to saponins found in *E. nobilis*. Thioglycoside chemistry was used for glycosylation reactions, with activation accomplished using NIS in the presence of La(OTf)_3_.

Okada et al. have found two unique 14-nor-methyl-23,24,25,26,27-pentanorlanostane glycosides in *Erylus placenta* harvested near the shores of the South Japan (Hachijo Island). An unprecedented sokodoside A (**71**) aglycone possesses an 8(14)-double bond. However, sokodoside B (**72**) has an unprecedented conjugated 8(9),14(15),16(17)-triene system [43] (Figure 6). Sokodoside **71** is a branched tetraoside. The first monosaccharide residue of **71** is β-D-galacturonic acid, the second sugar linked to C-2 of the first monosaccharide residue is also β-D-galacturonic acid, and the terminal (third) one linked to C-2 of the second monosaccharide residue is α-L-fucose. Another terminal monosaccharide is α-L-arabinose residue. Sokodoside B (**72**) is a branched trioside that has carbohydrate moiety including α-L-arabinose, β-D-galactose and β-D-galacturonic acid. The arabinose residues configuration was erroneously determined as β. However, the presented coupling constant in the ^1^H NMR spectra of **71** and **72** demonstrated an α-configuration. A synthesis of the trisaccharide carbohydrate chain of sokodoside B has been realized by thioglycoside activation using sulfuric acid immobilized on silica in conjunction with *N*-iodosuccinimide [59]. The glycosides **71** and **72** revealed moderate growth-inhibitory activity on the fungus *Mortierella ramanniana* and several strains of the yeast *Saccharomyces cerevisiae* with or without mutations (cdc28, erg6 and act1-1). The size of inhibition zones was varied from 8 to 16 mm at 50 μg of the tested glycoside on 6 mm spot on thin paper disk. The glycoside **71** was more active than **72**. Glycosides **71** and **72** reveal moderate cytotoxic effects on P388 cells with IC_50_ of 100 and 50 μg/mL, respectively. The correlation between their antifungal and cytotoxic activities was observed [43].

Thus, all the studied representatives of the genus *Erylus* contain bioactive tetracyclic triterpene glycosides, preferably having lanostane aglycones with C-14 carboxyl and carbohydrate chains with predominance of arabinose and galactose monosaccharide residues.

### 2.2. The Order Bubarida

A Pacific Ocean sponge *Lipastrotethya* sp. belongs to the family Dictyonellidae. It was harvested from the shallow waters of Chuuk Lagoon (Truk Lagoon), Micronesia. Five new triterpene glycosides: sarasinosides N−R (**73**−**77**) (Figure 7) [44] were isolated from this species along with known sarasinosides A_1_ (**1**), A_2_ (**2**) [21,24], H_1_ (**14**), H_2_ (**15**) [26], J (**18**), and M (**21**) [23].

All the aglycones of **73**–**77** are 14-nor-lanostane derivatives having a 23-ketone and a 24(25)-double bond in the side chain. The aglycone of **73** has an 8(9)-double bond whereas aglycones of **74** and **75** possess a 9(11)-double bond. Aglycone of **74** has additional 5α- and 8α-hydroxyls whereas aglycon of **75** possess 5β- and 8β-hydroxyls. The aglycone of **76** has an 8(14)-double bond and 9α-hydroxy and 8(9)-epoxy groups. The aglycone of **77** has 8α,9α-diol group and C-18 methyl shifted to 14β-position. It also has a unique 12(13)-double bond. Most of these glycosides feature a pentasaccharide chain that is very typical of sarasinosides. However, glycoside **73** is unique in that it is a trioside lacking two glucose residues, specifically the terminal and pre-terminal ones [44].

All the studied glycosides, including known ones, had no antibacterial action on several bacterial strains [44]. However, the most of glycosides possessed moderate cytotoxic activities on K562 leukemia and A549 lung carcinoma cell lines. Nevertheless, glycoside **77** was not active against K562 tumor cell lines whereas glycoside **15** was not active against A549 cells. The author noted that the presence of additional hydroxyls at the nucleus of an aglycone decreased the activity. Sarasinoside A_1_ (**1**), its derivative obtained from **1** by glycolysis, and sarasinoside N (**73**) weakly inhibited Na^+^/K^+^-ATPase with IC_50_ 60.0, 59.4, and 54.1 μg/mL, respectively [44].

In a continuation of this research, eleven triterpene glycosides belonging to the sarasinosides group, including a newly discovered glycoside named “glycoside 1” (**78**) (Figure 7), have been isolated from *Lipastrotethya* sp. [45]. The aglycone was a 14-nor-lanostane derivative with an 8(9)-double bond, and 23-keto and C-24 methylene groups in the side chain. Known sarasinosides A_1_ (**1**) A_3_ (**3**), B_2_ (**5**) [21,24], M (**21**) [23], A_4_ (**22**) [27], H_2_ (**15**), I_1_ (**16**), and I_2_ (**17**) [26], Q (**76**) and R (**77**) [44] were also isolated from this sponge. The structural formula of **78** and the formulae of relative compounds were presented in original article with error in carbohydrate chain where terminal 2-N-acetylgalactosamine was replaced with 2-N-acetylglucosamine and incorrect numbering in the side chain (double bond should be 24(28), not 24(25) as in the article). Here, we present the corrected formula of **78**. The cytotoxicity of the isolated compounds against four tumor cell lines was studied, and it was found that glycoside **78** was cytotoxic against several tumor cell lines, including ACHN (IC50 = 7.52 µg/mL), MDA-MB-231 (IC50 = 10.61 µg/mL), NCI-H23 (IC50 = 10.85 µg/mL), and NUGC-3 (IC50 = 10.47 µg/mL) [45].

A group of French investigators has isolated new triterpene bioside, eryloside W (**79**), from the sponge *Dictyonella marsilii* (family Dictyonellidae) harvested in the Gibraltar Strait (Figure 8). This is the first sponge triterpene glycoside found in the Mediterranean region. The glycoside has a lanostane aglycone with an 8(9)-double bond, a carboxyl group at C-14, a methylene group at C-24 in the side chain, and a carbohydrate chain consisting of β-D-glucuronic acid as the first sugar and 2-NAc-β-D-glucopyranose as a terminal monosaccharide residue linked with C-2 of the first glucose residue [46].

### 2.3. The Order Haplosclerida

Maarisit et al. have isolated a new 14,25,26,27-tetra-norlanostane triterpene glycoside, sarasinoside S (**80**), along with three known glycosides of the same class—sarasinosides A_1_ (**1**) [21,24], I_1_ (**16**) [26], and J (**18**) [23], which have the same carbohydrate chain but different aglycones, from an Indonesian marine sponge *Petrosia* sp. (family Petrosiidae) (Figure 9) [47]. The pentaoside carbohydrate chain is common for many sarasinosides, but the aglycone of this compound has an original side chain lacking carbons 25, 26, and 27 and a C-23 keto group.

The structure of the glycosides was elucidated using modern 2D NMR and HRMS procedures. The absolute configuration of **80** was proposed along with biogenetical reasons. All the isolated triterpene glycosides were not active against two human solid cancer cell lines, Huh-7 (hepatocarcinoma) and A549 (lung carcinoma) [47].

Two more glycosides with a similar carbohydrate chain, named as 5,8-epoxysarasinoside (**81**) and 8,9-epoxysarasinoside (82), with known sarasinosides A_1_ (**1**) [13,16], H_1_ (**14**), I_1_ (**16**), I_2_ (**17**) [18], O–R (**74**–**77**) [38] were isolated recently from the marine sponge *Petrosia nigricans* [48]. Glycoside **81** possesses an epoxy group at positions 5 and 8, whereas glycoside 82 possesses an epoxy group at positions 8 and 9. Furthermore, compound **81** has a double bond at position 9(11), whereas compound **82** has a double bond at position 12(13) and a methyl group at position 14β instead of a methyl at position 18, similar to sarasinoside R (77). Glycosides **81** and **82** have shown low cytotoxicity against human colon cancer HCT116 and lung cancer A549 cell lines [48].

### 2.4. The Order Poecilosclerida

A sponge *Ulosa* sp. from the Indian Ocean belongs to the family Esperiopsidae. Antonov et al. found ulososides A–E (**83**–**87**), five new glycosides with oxidized 14-nor-methyl-lanostane aglycones. This sponge was collected shallow waters of North-Western Madagascar [49,50,51]. Ulososides A (**83**) [49], C (**85**), D (**86**), and E (**87**) [51] are characterized by the presence of a carboxyl group at C-4, a 22S,23R-diol fragment, and a 24S-methyl group in the side chains of their aglycones. These structural features are similar to those found in plant hormones known as brassinolides [60]. Ulososide B (**84**) possesses an aglycone with both oxidized methyl groups at C-4 to carboxyl and hydroxymethyl groups, respectively, with 23ξ-hydroxyl in the side chain, but it has no a 24-methyl [50]. Compound **83** is a bioside that has a rare 1→6 interglycosidic bond between D-glucose and terminal D-glucuronic acid residues. However, glycosides **84**–**86** are monosides with N-acetyl-2-deoxy-2-amino-β-D-glucose or D-glucose as a carbohydrate part. Similarly to glycoside **83**, ulososide E (**87**) is a bioside, but its carbohydrate chain possesses a different monosaccharide composition, having glucuronic acid and uncommon terminal monosaccharide residue—α-D-arabinopyranose (Figure 10).

The sponge *Ectyoplasia ferox* is a Caribbean species of sponges of the family Raspailiidae. Ectyoplaside A (**88**) and B (**89**), as well as feroxosides A (**90**) and B (**91**), four new 4β-nor-lanostane triterpenoid glycosides, containing aglycones with oxidized methyls at C-4, have been found in the glycoside fraction from two samples of this sponge harvested in the shallow waters of the Bahamas [52,53]. Glycosides **88** and **89** are linear triosides, with two α-L-arabinopyranose and β-D-galactopyranose residues. Feroxosides **90** and **91** are branched at the first sugar tetraosides, having two residues of β-D-glucopyranose and two residues of α-L-rhamnopyranose. Residues of α-L-rhamnose were never discovered in sponge triterpene glycosides before this investigation (Figure 11). Glycosides **88** and **89** reveal moderate cytotoxic effects in vitro on J774 (murine monocyte-macrophage), WEHI164 (murine fibrosarcoma), and P388 (murine leukemia) cell lines (IC_50_ from 8.5 to 11.4 μg/mL) [52]. Glycosides **90** and **91** were moderately cytotoxic (IC_50_ of 19 μg/mL) against the cells of J774 murine monocyte-macrophages [53].

A group of authors from Colombia and France have reported the isolation of three new triterpene glycosides, namely ulososide F (**92**), urabosides A (**93**) and B (**94**) (Figure 11), from a sample of the Caribbean marine sponge *Ectyoplasia ferox*. The previously known ulososide A (**83**) was also isolated in this study. The sponge was harvested in the Uraba Gulf (Colombia). All the compounds have an aglycone derived from 14-normethyl lanostane with a double bond at the 8(9)-position. The side chain of **92** composed 22α- and 23β-hydroxyls and additional 24β-methyl. The carbohydrate chain includes 2-NHAc-β-D-glucopyranosyl as the first monosaccharide residue and β-D-glucuronic acid as the terminal monosaccharide residue attached to C-6 of the first sugar. The aglycones of urabosides A (**93**) and B (**94**) are 14-normethyl derivatives with an 8(9)-double bond and a 23-ketogroup. Both aglycones have a 4β-methyl group that is oxidized by a carboxyl group. The aglycone of **93** has a 4α-methyl group that is oxidized by CH_2_OH, whereas the aglycone of **94** has a 4α-methyl group that is oxidized by a carboxyl group.

The first monosaccharide residue of the triosidic carbohydrate chain in glycoside **93** is branched, and it is a β-D-galactopyranose. The terminal monosaccharide residues, β-D-arabinopyranose and β-D-galactopyranose, are linked to C-2 and C-3 of the first monosaccharide residue, respectively. There is no significant cytotoxicity on the two cell lines (Jurkat and CHO cells) or hemolytic action for the isolated compounds [54].

In order to estimate of the diversity of triterpene glycosides in the same collection of *E. ferox*, the authors used an metabolomic approach and applied the HPLC-ESI-IT-MS/MS method [61]. They have obtained valuable information about the presence of 25 compounds including three that were previously reported and three which are a combination of known aglycones and different carbohydrate chains. The saponins mixture revealed a significant cytotoxic effect on the CHO-k1 and Jurkat cell lines, as was found by the MTT test [61].

It can be noted that most of the glycosides isolated from the sponges of the order Poecilosclerida have aglycones with an 8(9)-double bond and one or two oxidized methyl groups at C-4. The carbohydrate chains of these compounds range from one to four monosaccharide residues. Unusual monosaccharides, as well as α-D-arabinopyranose, β-D-arabinopyranose, and α-L-ramnopyranose, were found.

The isolation of a series of holostane triterpene glycosides from the sponge *Ianthella basta* (family Iantellidae, order Virongiida) collected in Nha Trang Bay (Vietnam) was reported. These glycosides are very characteristic of sea cucumbers of the family Holothuriidae in terms of both aglycone and carbohydrate chain structures. The isolated compounds include known holothurin A_2_, desulfoechinoside A, echinoside B, holothurin A, holothurin B, and a new glycoside named lanthebastanoside A. However, there is some doubt regarding the isolation procedure as it is possible that the sea cucumber extract belonging to the family Holothuriidae was mixed with the sponge extract. The authors did not provide any explanation or clarification regarding this issue [62].

## 3. Steroid Glycosides from Sponges

### 3.1. The Order Tetractinellida

Pachastrelloside A (**95**) is a steroid bioside obtained from a sponge of the species Pachastrella sp. (belonging to the suborder Astrophorina and family Pachastrellidae). Its sterol aglycone contains a 5(6)-double bond and is oxidized at positions C-2, C-3, C-4, and C-7 in rings A and B. The aglycone is also attached to two sugars, namely D-galactopyranose and 4-O-acetyl-β-D-xylopyranose, at positions C-4 and C-7 (as depicted in Figure 12). [63]. This glycoside inhibits cell division of the fertilized eggs of the starfish *Asterina pectinifera*. However, it does not affect nuclear divisions to form multinucleated, unicellular embryos.

Scrobiculosides A (**96**) and B (**97**), two new steroid glycosides, have been found in the deep-sea sponge *Pachastrella scrobiculosa* which is harvested near the coast of Miura Peninsula, Japan (Figure 12). The aglycones of scrobiculosides A and B possess a saturated nucleus which is oxidated to carboxyl C-18 and have a unique vinylic cyclopropane and exomethylene group at C-24 in the side chains, respectively. These saponins have bioside carbohydrate chain with β-D-galactopyranosyl as the first sugar and the linked to C-2 of the first monosaccharide residue β-D-glucopyranoside, whereas glycoside **96** has an acetyl group on C-6 of the glucose residue. Glycoside **96** revealed cytotoxicity on P388 and HL-60 tumor cell lines (IC_50_ = 61 and 52 μM, respectively) [64].

A two-sponge symbiotic association was also studied by Korean authors. This association included *Poecillastra wondoensis* (family Vulcanellidae) and *Jaspis wondoensis* (family Ancorinidae (*Jaspis wondoensis* is the unaccepted name, the valid name is *Rhabdastrella wondoensis*)). The sponge association was harvested near Cheju Island, Republic of Korea. Wondosterols A–C (**98**–**100**), three steroid glycosides, were found in this association. Their structures including absolute stereochemistry were elucidated by NMR spectroscopy and application of Mosher’s method (Figure 13). The structures of **98**–**100** are similar to **95**, although **98**–**100** contain only one carbohydrate chain linked to C-4 and hydroxyl at C-7 with β-configuration. The side chains of wondosterols A–C differ in the position of the double bond. Their carbohydrate moieties include D-xylose and D-galactose monosaccharide residues [65].

The deep-water sponge *Poecillastra compressa* (family Vulcaniidae) harvested near French coasts of Mediterranean Sea was studied by a French group of scientists. They isolated seven new steroid glycosides, poecillastrosides A–G (**101**–**107**) (Figure 13) [66]. All the glycosides are characterized by oxidized C-18 methyl by a primary alcohol or carboxylic acid. Poecillastrosides A–D (**101**–**104**) contain a methylene or ethylene group at C-24 of the side chain and the carbohydrate chain consists of the first β-D-galactopyranosyl residue and is linked to C-2 β-D-glucopyranosyl residue. Poecillastrosides E–G (**105**–**107**) have aglycones with a 22(23)-E-double bond and a 24(25)-cyclopropane ring in the side chain, which is similar to scrobiculoside A (**96**) [64]. Additionally, the glycosides have two identical monosaccharide residues linked to each other through C-3 of the first sugar. However, the terminal glucose monosaccharide residues are acetylated by C-6 in the glycosides **105** and **107**. It was found that poecillastrosides D and E, having a carboxylic acid at C-18, possess antifungal activity against *Aspergillus fumigatus* (MIC_90_ = 6 μg/mL and 24 μg/mL, respectively) [66].

### 3.2. The Order Poecilosclerida

The sample of Caribbean sponge *Pandaros acanthifolium* belonging to the family Microcionidae was harvested from canyon rock near Martinique Island. A series of new steroid glycosides, named pandarosides, were isolated as result of the investigation into this sponge. Pandarosides A–D (**108**–**111**) are glycosides with uncommon sterol aglycones, which have an unusual cis-C/D ring junction that is oxidized in ring D [67]. Their carbohydrate chains include the residues of D-glucose and D-glucuronic acids or D-glucuronic acid. The carbohydrate moiety is attached to C-3 of the aglycone (Figure 14). During the isolation procedure, methyl esters 108a, 110a, and 111a of pandarosides A, C, and D, respectively, were obtained. It is likely that these methyl esters are artefacts of a chemical reaction with methanol using as an extractant. The absolute configuration of the aglycone moiety of 108 was determined by comparing the experimental and TDDFT-calculated CD spectra of the more stable conformer. All the isolated glycosides showed the absence of the activity on three human tumor cell lines (MDA-MB-231 breast cancer cells, HT29 colonic cancer cells, and A549 lung cancer cells) below 10 μg/mL [67].

In continuation, the next series of steroidal glycosides, pandarosides E–J (**112**–**117**) and their methyl esters **112a**–**117a**, have been isolated from the same sponge [68] (Figure 14). These compounds were tested for in vitro antiprotozoal activity against four parasitic protozoa and cytotoxic activity on L6 cancer cells. Pandaroside G (**114**) and its methyl ester **114a** were inhibitors of the growth of *Leishmania donovani* (IC_50_ of 1.3 and 0.051 μM, respectively) and *Trypanosoma brucei rhodesiense* (IC_50_ of 0.78 and 0.038 μM, respectively) [68].

Finally, pandarosides K–M (**118**–**120**) were isolated as minor components from the same species [69] and their methyl esters **118a**–**120a** were also obtained during the isolation procedure probably as artefacts (Figure 14). All these metabolites and isolated early compounds **114** and **114a** were moderate inhibitors of the growth of four parasitic protozoa and did not indicate cytotoxicity on mammalian cells, except for pandaroside G (**114**) and its methyl ester (**114a**), which were inactive on *T. b. rhodesiense* [69].

Thus, all pandarosides have rare aglycones with 14β-H, a 16(17)-double bond, and 3β,16-diol and 15-keto groups. Pandarosides E and F (**112**, **113**) and pandarosides G and L (**114**, **119**) have additional 8(9)- and 7(8)-double bonds, respectively. All pandarosides have a 23-keto group in their side chains. Differences were observed at C-24, which is either free or contains methyl, ethyl, methylene, or ethylene groups. Four types of carbohydrate chains were found in pandarosides, and all contain β-D-glucuronic acid as the first monosaccharide residue. The obtaining methyl esters of pandarosides, most likely, are artifacts formed during the MeOH extraction steps.

Another series of steroid glycosides from the same species are acanthifoliosides A–F (**121**–**126**) (Figure 15). Acanthifoliosides are closed to pandarosides steroid saponins. However, they have common steroid nuclei with a *trans*-junction of C and D rings. Similar to aglycones of pandarosides, the aglycones of **121**–**126** possess an oxidized functionality in the D ring. The configurations of C-23 and C-24 centers in the side chains of **124**–**126** were not established because the low amounts of substances for using Mosher’s method. Although some assumptions about relative stereochemistry of the side chains have been made, the authors noted that confirmation of configurations C-23 and C-24 required more thorough stereochemical analysis.

The carbohydrate residues of acanthifoliosides A–F (**121**–**126**) are connected to either C-15 (in glycosides **121**–**123**) or C-16 (in glycosides **124**–**126)** of their steroid aglycones. Acanthifoliosides A–C (**121**–**123**) are mono-β-D-xylopyranosides, whereas acanthifoliosides D and E (**124** and **125**) are mono-α-L-rhamnopyranosides. However, acanthifolioside F (**126**) is a branched trioside (Figure 15). The methyl ester of acanthifolioside F (**126a**) was also isolated and probably formed during the extraction process as previously observed for pandarosides [67,68,69]. Moderate antiprotozoal activity has been reported for acanthifoliosides A–F (**121**–**126**) and the methyl ester of acanthifolioside F (**126a**) [70]. There is a definite structural closeness of *P. acanthifolium* glycosides and starfish steroid glycosides, because of oxidation of D-ring in C-15 and C-16 positions [16,17,18,19].

The next sample of *P. acanthifolium* was harvested through scuba diving near the coast of Marathon in the Florida Keys (FL, USA) and was investigated by a group of Canadian chemists and biochemists [71]. They have isolated four new minor steroid glycosides, acanthifoliosides G–J (**127**–**130**) (Figure 15). These glycosides also were characterized by a highly oxygenated D ring. All the isolated glycosides possess β-L-glucopyranose as a first sugar with a terminal α-L-rhamnopyranosyl unit linked to C-2 of the first sugar. This bioside carbohydrate chain attached to C-3 of steroid aglycone whereas another carbohydrate moiety composing of α-L-rhamnopyranosyl residue is attached to C-15 aglycone. The absolute configurations of glucose and rhamnose were found by aldose o-tolylthiocarbamate derivatization followed by comparison with reference substances by LC/HRESIMS. The aglycones of **127**, **128** are 5α-cholestane derivatives with 16β-*O*-acetic group. The aglycone of **128** has a 22(23)*E*-double bond in the side chain. The aglycones of **129** and **130** are 5α-poriferastan and poriferast-5-en derivatives, respectively. The side chains of the both of aglycones possess a hydroxyl at C-23 and ethyl group at a 24-position, as well as in acanthifoliosides D–F [70]. The hydroxyl configuration was not found because of too low amounts of the substances. The glycosides were tested on antioxidant and cell-protective activities, but only acanthifolioside G (**127**) exhibited antioxidant and cytoprotective activities [71].

The steroid oligoglycosides, similar to starfish asterosaponins by containing several monosaccharide residues, were first found in a sponge of the genera *Mycale* (the family Mycalidae) by Russian scientists in 1981 [72]. Unfortunately, the separation of the corresponded glycosidic fractions was impossible because of a low level of development of isolation technologies. Later, steroid oligoglycoside mycaloside A (**131**) was isolated from the Caribbean sponge *Mycale laxissima* as individual substance [73]. Then, the structures of related mycalosides B–K (**132**–**141**) were reported [74,75] (Figure 16). All the isolated glycosides were tetraosides with the similar carbohydrate parts architecture. They include two D-galactopyranose, one D-glucopyranose, and one D-arabinopyranose residues. In mycalosides D, F, and K (**134**, **136**, **141**) the glucopyranose residue has acetate group at C-6. One galactopyranosyl moiety attached to C-4 of glucopyranosyl residue by a β-glycosidic bond, another galactopyranosyl unit attached to C-2 of arabinopyranose by a α-glycosidic bond. Their aglycones are steroid derivatives, as they have oxidized functionalities in rings A, D, and side chains. All mycalosides, with the exception of mycaloside I (139), have an unusual oxidation at C-21, which is characteristic of steroids from ophiuroids [16,17]. Mycalosides B (**132**) and C (**133**) are 27- and 28-norsteroid derivatives of the glycoside **131**, respectively. Mycaloside D (**134**) differs from mycaloside A (**131**) only by the presence of an additional acetyl at C-6 of the first glucose in the carbohydrate moiety. Mycaloside E (**135**) is a 28-nor-4-deoxy-mycaloside A. Mycalosides F–H (**136**–**138**) have new 5(6)-unsaturated 3β,4β,21-trihydroxy-15-keto-steroidal aglycones and differ from each other by aglycone side chains. These glycosides have nonacetylated (**137**, **138**) or acetylated by C-6 of the first glucose residue (**136**) in the tetrasaccharide carbohydrate chains. Mycaloside I **(139**) contains a tetraoside with a novel aglycone featuring a 7,24(28)-diunsaturated-3β,15β,29-trihydroxystigmastane derivative. Mycaloside I (**135**) contains a new aglycone, which differs from the aglycone of **131** by the absence of C-4 hydroxyl. Mycaloside K (**141**) is C-24 epimer of mycaloside D (**134**) (Figure 16). The total fraction of the mycalosides and individual mycalosides A (**131**) as well as micaloside G (**137**) have an inhibition action on the fertilization of eggs by sperm of the sea urchin *Strongylocentrotus nudus* after preincubation [74].

### 3.3. The Order Haplosclerida

The sponge *Niphates olemda* (formerly known as *Cribrochalina olemda*), (family Niphatidae) is characteristic for the Indo-Pacific region and was collected at a depth of 40 m in Micronesia. The found in this sponge hapaioside (**142**) has an uncommon aglycone with a polycyclic nucleus, resembling several steroid hormones. It has a 4-hydroxy-6-oxo-19-norpregnane skeleton. The glycoside is a monoside and its monosaccharide residue is 6-deoxy-β-L-altropyranose-4-acetate [76] (Figure 17). This sugar was earlier found as a constituent of a lipopolysaccharide isolated from mammalian intestinal microorganisms.

### 3.4. The Order Axinellida

*Ptilocaulis spiculifer* is a sponge belonging to the family Axinellidae and is found in the tropical waters of the Indo-Pacific region. This sponge is known to contain sulfated polyhydroxysteroids. During the isolation procedure of this sponge specimen, two pregnane glycosides, ptilosaponosides A (**143**) and B (**144**), as well as sulfated polyhydroxysteroids were also obtained. The specimen was collected from shallow waters in the Solomon Islands. (Figure 17). The glycosides contain β-D-glucopyranosyl-3-*O*-sulfate residue as a carbohydrate chain and their rings A are oxidized at C-3, C-4, and C-19. The glycoside **143** also has an additional sulfate in the aglycone. The absolute configuration of glucose was not assigned due to the small amount of isolated compounds. Glycosides **143** and **144** have no cytotoxic action on human tumor KB cells [77].

Thus, steroid glycosides of marine sponge are very interesting class of marine natural products. Some of them also reveal significant antiprotozoal activity and other kinds of biological activity.

## 4. Taxonomic Distribution of Glycosides in Sponges

Frequently, the presence of occurring triterpene and steroid glycosides in sponges can result in highly complex mixtures, making their separation a challenging task. Consequently, the difficulty in isolating these glycosides may hinder their utilization as chemotaxonomic markers. Due to the complexity of glycoside fractions, many researchers may only isolate a portion of the components at random, whereas another group may isolate a different portion. Comparing such data may lead to inaccuracies. Furthermore, distinct structural series of triterpene and steroid glycosides are distributed among taxa that are different from each other and widely separated. The most frequently the tetracyclic triterpene and steroid glycosides are presented in two orders—Tetractinellida (suborder Astrophorina) and Poecilosclerida. In total, the glycosides were found in the sponges belonging to five orders (Table 1 and Table 2, the taxonomic names corresponded to WoRMS—World Register of Marine Species data base [78]). The distribution of glycosides occurs across multiple families and species and is not strictly dependent on ecological or geographical factors. The major glycoside in *Melophlus sarasinorum* is sarasinoside A_1_ (**1**), which has been harvested in various geographical locations ranging from Guam to the north-western coast of Australia. Sarasinoside A_3_ (**3**) has been isolated from *M. sarasinorum* harvested in shallow waters near the north-western coast of Australia, Guam Island, Palau Islands, and Sulawesi Island. Similarly, sarasinoside A_2_ (**2**) was found in specimens harvested from shallow waters near the north-western coast of Australia, Palau Island, and other regions. A very similar character of distribution was found for erylosides of *Erylus formosus*, from which formoside (**47**) and eryloside F (**49**) were isolated from the samples harvested near Puerto Morelos and Bahamas in the Gulf of Mexico. Discoveries of glycosides with identical structures in different species, even those within the same genus, are uncommon. However, an instance of such a finding is eryloside H (**67**), which has been identified in *Erylus formosus* harvested near Puerto Morelos and in samples of *E. nobilis* from the waters of Jaeju Island, Republic of Korea. Nevertheless, some sponge genera may be characterized by the occurrence of closely related glycosides. This may be exemplified by erylosides found in different species of *Erylus*, which was collected in the shallow waters of three oceans (Pacific, Atlantic and Indian). These glycosides contain 14-carboxy- or 14-nor-methyl-lanostane aglycone. Some of them are additionally dealkylated at C-4 and alkylated in sidechains. General carbohydrate chains architectures for a series of glycosides isolated from *Erylus* spp. are frequently close, but, as a rule, exact structures are a little different.

As we already have noted, a Korean group recently reported on the finding of a series of sarasinosides from the sponge *Lipastrotethya* sp. belonging to the order Bubarida, the family Dictyonellidae, which was harvested in Micronesian shallow waters [44]. Another representative of this order, *Dictyonella marsilii* contains eryloside W (**79**) very similar to other glycosides of eryloside series [46]. A series of sarasinosides was also isolated from *Petrosia* sp. and *Petrosia nigricans*, the representative of the order Haplosclerida [47,48]. Very similar steroid glycosides containing unique aglycone side chains having a 23(23)-E-double bond along with a 24(26)-cyclopropanic ring were isolated from representatives of two different families of the order Tetractinellida (suborder Astrophorina)—*Pachastrella scrobiculosa* (family Pachastrellidae) and *Poecillastra compressa* (family Vulcanellidae), which were harvested near Miura Peninsula, Japan [64] and the French coasts of Mediterranean Sea [66], respectively.

The discovery of similar occurrences suggests a parallel origin and evolution of terpenoid and steroid glycosides in Demospongiae.

Therefore, triterpene and steroid glycosides can be regarded as taxonomic markers that are specific to certain species and occasionally even to the level of genus or subgenus. However, the use of them for improvement of sponge taxonomy may be difficult because of the presence of very complicated mixtures of close related substances in glycosidic fractions, and seasonal and ecological variability in the content of some components of glycosidic fractions in the different collection of the same sponge. This was described, for example, by Kubanek et al. [32], who found the presence of formoside in *Erylus formosus* harvested near Bahamas and its absence in the specimens collected in Floridian waters.

## 5. Biological Roles of Sponge Triterpene and Steroid Glycosides

Sponges are very important parts of coral reef ecological systems [79]. Kubanek et al. [80] discovered that triterpene glycosides from *Erylus formosus* harvested in shallow waters near Bahamas and Southern Florida deters fish-predators *Thalassoma bifasciatum*. Formoside (**47**) and formoside B (**48**), which have four monosaccharide residues in their carbohydrate chains, were found to be less active compared to the non-separated hexaoside fraction, which contains penasterol as the aglycone. The total glycoside sum indicated more activity than any separate subfraction or individual glycoside. Thus, triterpene glycosides of sponges may be a defensive agent against fish-predators [81].

The investigation on triterpene glycosides from two Caribbean sponges *Erylus formosus* and *Ectyoplasia ferox* demonstrated the activities of the glycosides not only as fish deterrents, but also as antifouling agents against biofilm-forming bacteria, invertebrates, and algae [82,83]. Moreover, they inhibited an overgrowth of neighboring sponges (allelopathy) in laboratory and field experiments. The multiple ecological functions of sponge triterpene glycosides have been elucidated [79,80,81,82,83,84,85]. An antifouling function may be caused by their antimicrobial properties, known for sarasinoside J (**18**) [23], sokodosides [43], some erylosides [29,31,41], and other triterpene glycosides. Strong antiprotozoal action, as, for example, for pandaroside G (**114**) [68], may also be a contribution into protection against pathogenic microorganisms and fouling. Allelopathic action of the glycosides may be caused by strong cytotoxic effects [22,26,41], including the action against embryos of organisms-competitors. Such activity is known for pachastrelloside A (**95**) [63], as well as for sarasinoside A_1_ (**1**) [22].

Ecological activities of sponge triterpene glycosides may be moderated even by minor structural differences. Some triterpene glycosides are not excreted in sea water, but their concentration on surface of sponge specimens was high. However, these sponges may deter their enemies and competitors by surface direct contacts with triterpene glycosides. This is more probable than by contacts via sea water [80]. The discovery of the molecular mechanisms of predator deterrence by triterpene glycosides in sponges also suggests their role as signaling substances, indicating their release into the surrounding seawater. It was discovered that zebrafish *Danio rerio* rejected an artificial diet containing the sponge triterpene glycosides. Transcripts from a zebra fish cDNA library were expressed in oocytes of *Xenopus laevis* and checked for activation of a chemoreceptor via electrophysiology and electrophysiological responses of ectyoplasides A, B (**86**, **87**), and formoside (**47**) were revealed [84]. Later, a new RAMP-like triterpene glycoside receptor (RL-TGR) was discovered in predatory fish [84,85]. It was involved in chemical signaling because this receptor responds to triterpene glycosides as chemical defensive substances in the water environment [85]. Hence, sponge triterpene glycosides may have antifouling and allelopathic roles.

Biological activities of sponge steroid glycosides are similar to such of terpenoid ones (see Table 1 and Table 2). For example, many steroid glycosides have been found to possess cytotoxic activity, suggesting their potential role as defensive agents against predators, as well as antifouling and allelopathic agents. However, the biological role of sponge steroid glycosides has not been specially studied.

## 6. Conclusions

About 144 sponge saponins were isolated by the end of 2022, including 94 tetracyclic triterpenoid glycosides and 50 steroid ones. The presence of triterpene and steroid glycosides is a characteristic feature of a few taxonomic groups (up to date, they were indicated in about 24 species, namely 15 species contain triterpene glycosides and 9 steroid ones). All of them were found within the class Demospongiae, but they are not characteristic for the class as a whole. The fact that terpenoid and steroid glycosides are distributed in a mosaic pattern within the class Demospongiae, and that glycosides with similar structures can be found in representatives of different orders, along with the data on their biological activities and roles, suggests that these substances may have evolved independently as defensive agents in several taxonomic groups of Demospongiae, and may have undergone parallel evolution.

These glycosides are very diverse from a chemical point of view and quite different from the structures of the both aglycone and carbohydrate chains. They are more structurally diverse than other saponins from marine invertebrates or terrestrial plants. The structures of both steroid and triterpene aglycones are the result of intensive oxidative processes that allow high activity of the corresponding cytochrome P-450 family of enzymes in the sponges belonging to saponin-producing taxa. These glycosides may be not only mono- and biosides, but also oligoglycosides. All the monosaccharides are in pyranose form and most of them belong to D-series; however, L-sugars also may be found. The monosaccharides may be acetylated by C-6 or may be in uronic acid form (glucuronic and galacturonic acids); however, they do not contain any OMe groups. Sponge triterpene glycosides demonstrate deterrent and ichthyotoxic properties, and many of them and their steroid analogs have cytotoxic and antiprotozoal activities that may be a cause of their defensive, antifouling, and allelopathic biological functions. It seems probable that new marine expeditionary research and the progress in development of separation and spectral technique may allow for the discovery of new, interesting structural versions of triterpene and steroid glycosides in sponges.

## Figures and Tables

**Figure 1 molecules-28-02503-f001:**
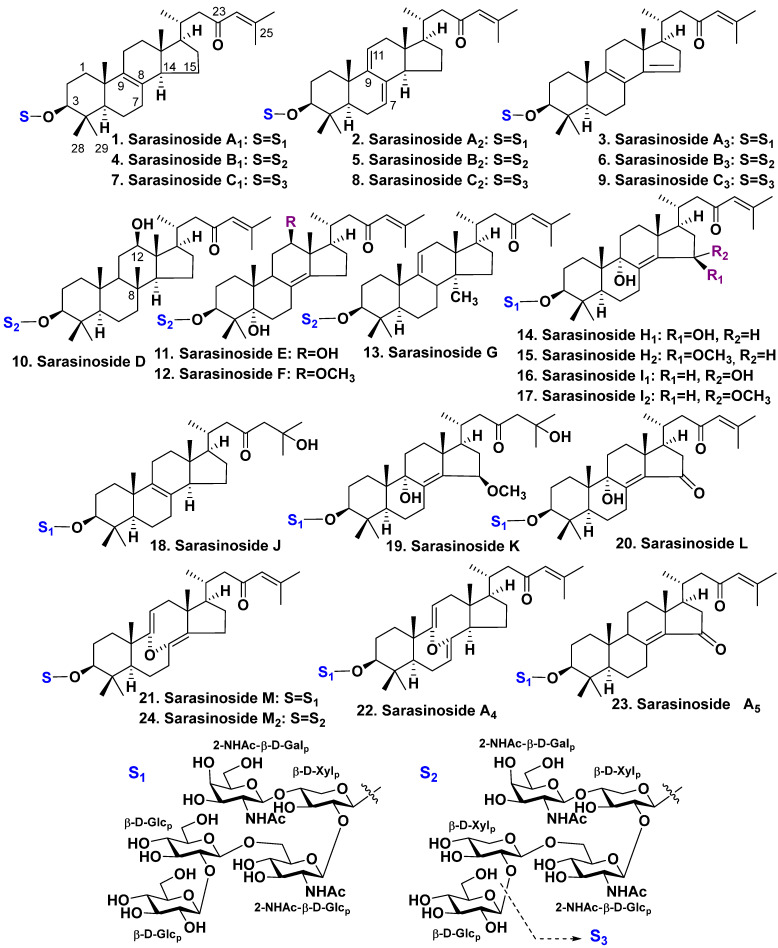
The structures of sarasinosides **1**–**24**.

**Figure 2 molecules-28-02503-f002:**
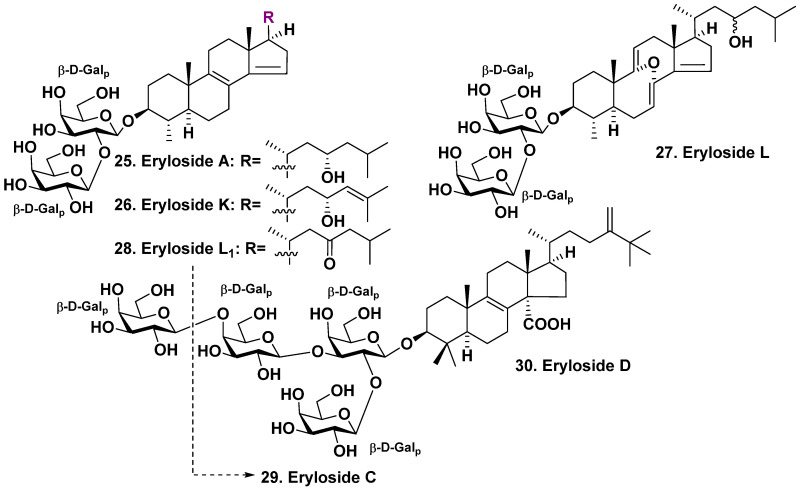
Erylosides **25**–**28** from *Erylus lendenfeldi* and erylosides **29**, **30** from *Erylus* sp.

**Figure 3 molecules-28-02503-f003:**
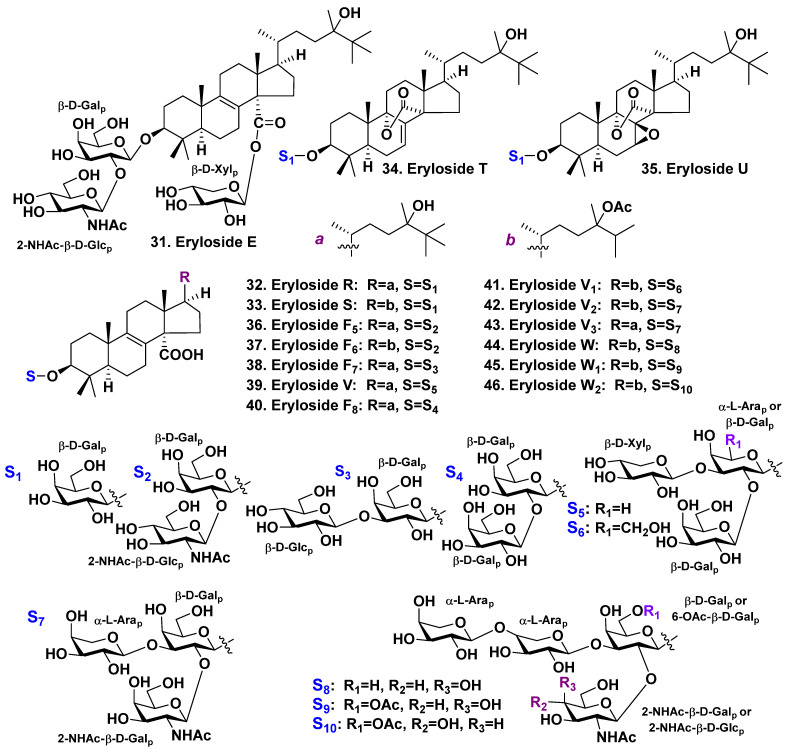
Erylosides **31**–**46** from *Erylus goffrilleri*.

**Figure 4 molecules-28-02503-f004:**
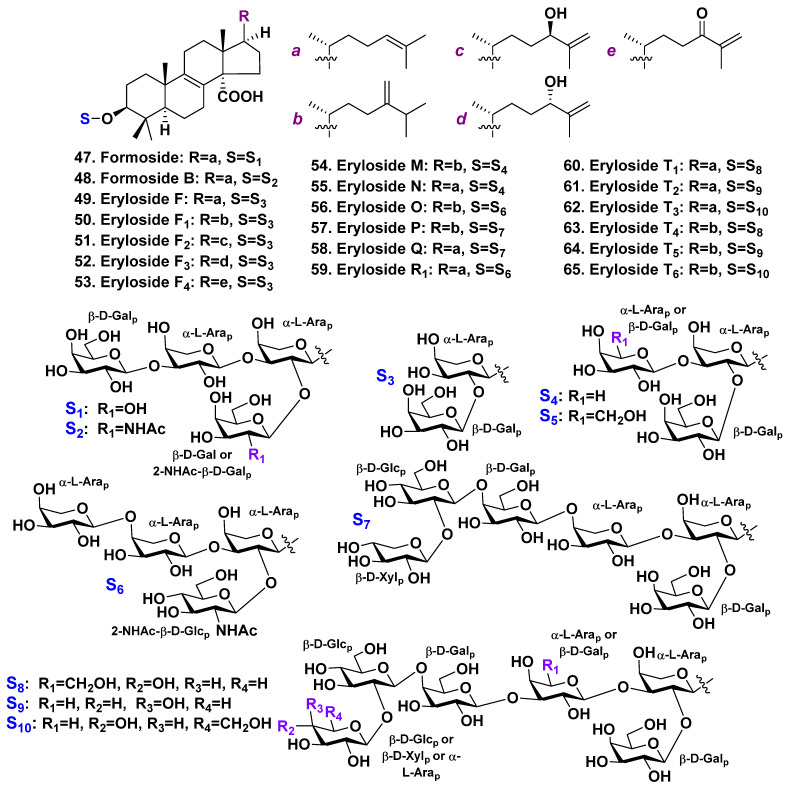
Erylosides **47**–**65** from *Erylus formosus*.

**Figure 5 molecules-28-02503-f005:**
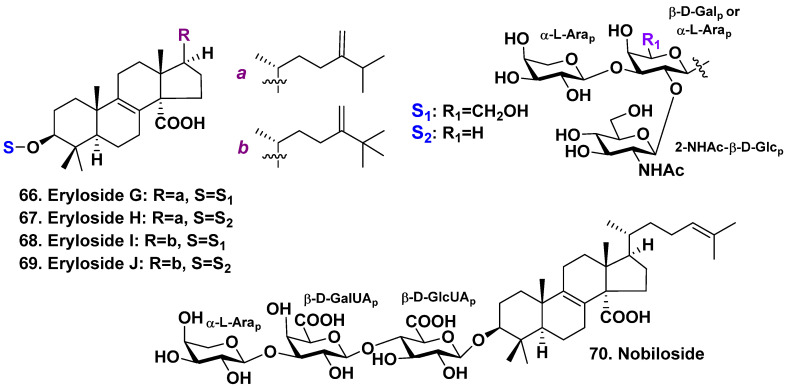
Erylosides **66**–**70** from *Erylus nobilis*.

**Figure 6 molecules-28-02503-f006:**
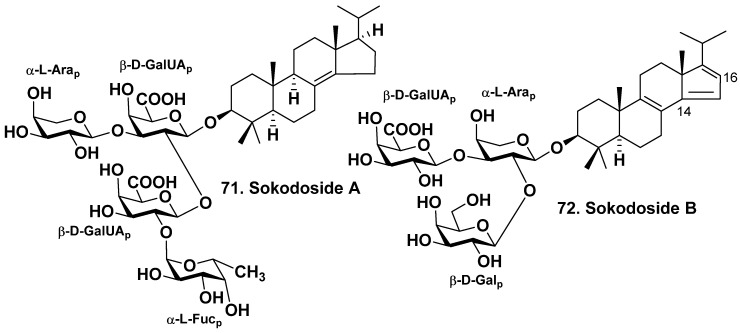
Sokodosides A (**71**) and B (**72**) from *Erylus placenta*.

**Figure 7 molecules-28-02503-f007:**
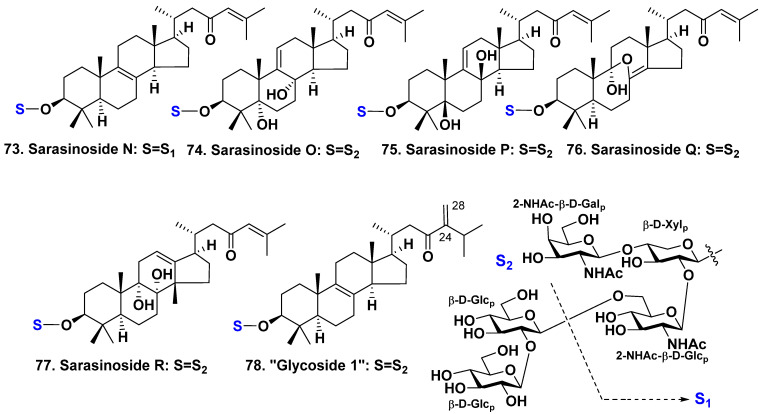
Structures of sarasinosides **73**–**77** and “glycoside 1” (**78**) from *Lipastrotethya* sp.

**Figure 8 molecules-28-02503-f008:**
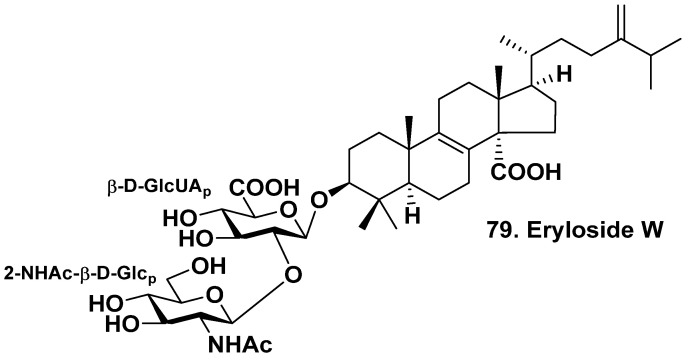
Eryloside W (**79**) from *Dictyonella marsilii*.

**Figure 9 molecules-28-02503-f009:**
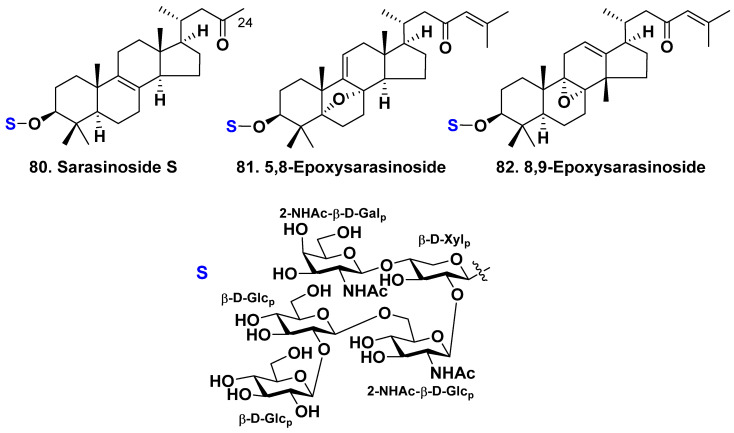
Sarasinoside S (**80**) from *Petrosia* sp. and 5,8-epoxysarasinoside (**81**) and 8,9-epoxysarasinoside (**82**) from *Petrosia nigricans*.

**Figure 10 molecules-28-02503-f010:**
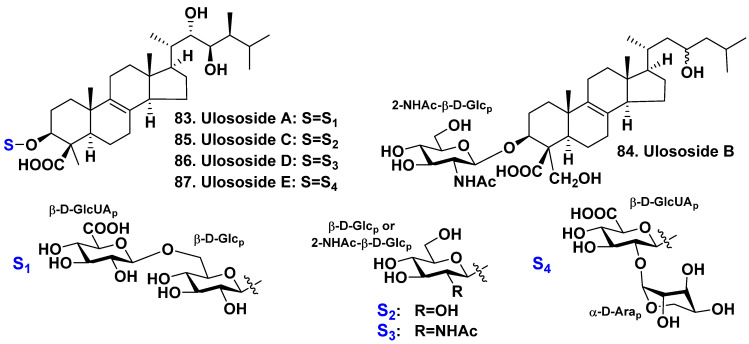
Ulososides **83**–**87** from *Ulosa* sp.

**Figure 11 molecules-28-02503-f011:**
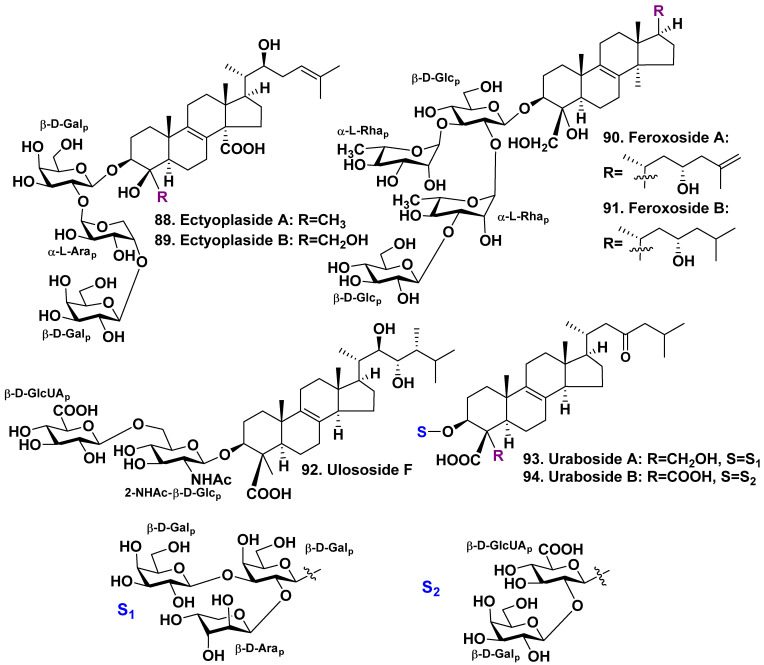
Glycosides **88**–**94** from *Ectyoplasia ferox*.

**Figure 12 molecules-28-02503-f012:**
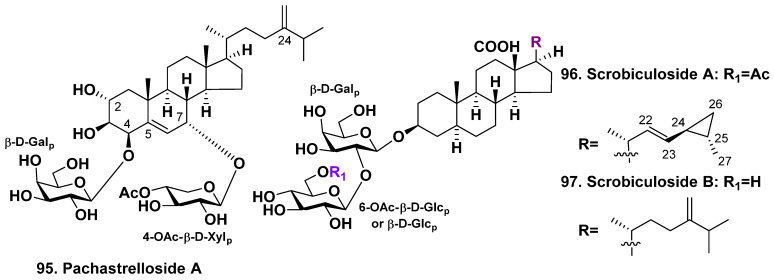
Structure of pachastrelloside A (**95**) from *Pachastrella* sp., and scrobiculosides A (**96**), and B (**97**) from *Pachastrella scrobiculosa*.

**Figure 13 molecules-28-02503-f013:**
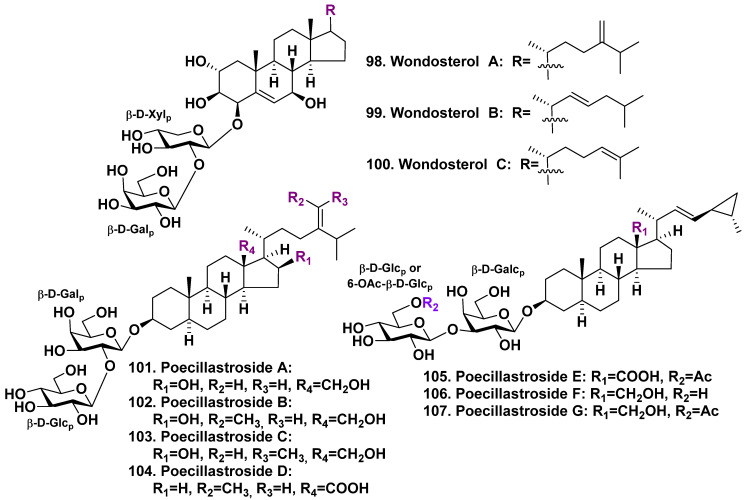
Wondosterols A–C (**98**–**100**) from a two-sponge association and poecillastrosides A–G (**101**–**107**) from *Poecillastra compressa*.

**Figure 14 molecules-28-02503-f014:**
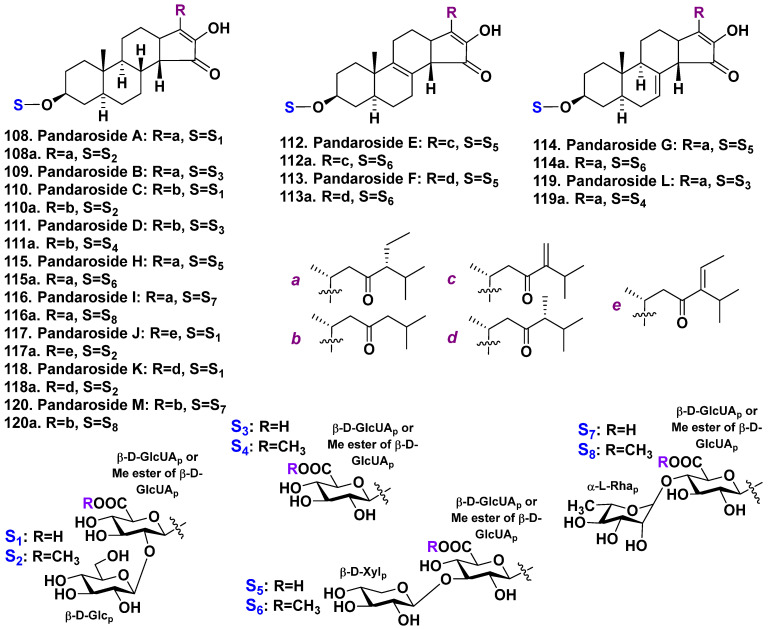
Pandarosides **108**–**120** and their methyl esters **108a**, **110a**–**120a** from *Pandaros acanthifolium*.

**Figure 15 molecules-28-02503-f015:**
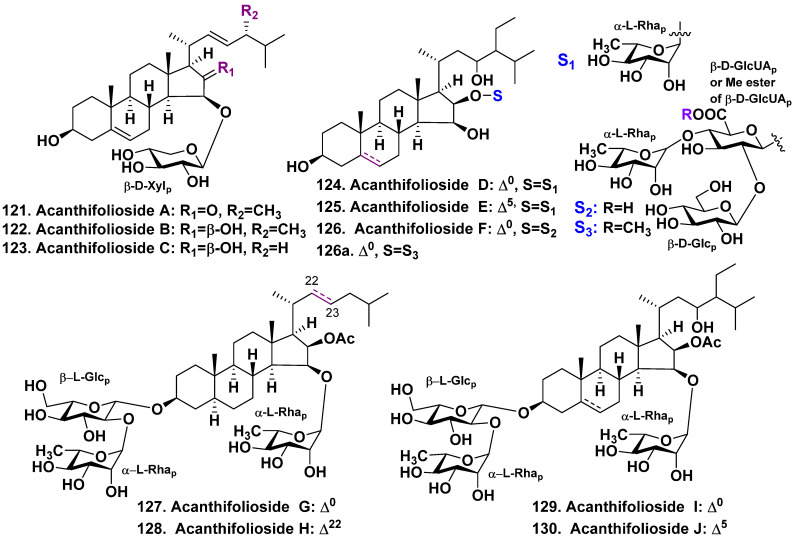
Acanthifoliosides **121**–**130** and methyl ester of acanthifolioside F (**126a**) from *Pandaros acanthifolium*.

**Figure 16 molecules-28-02503-f016:**
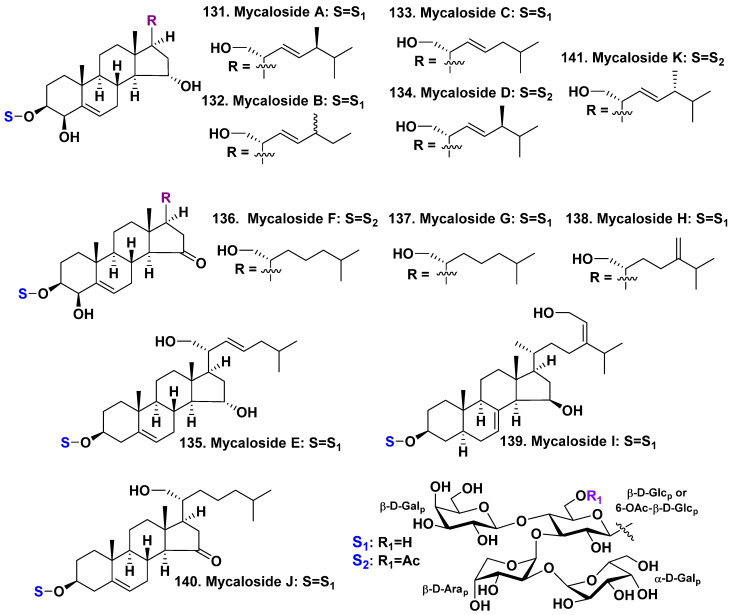
Mycalosides **131**–**141** from *Mycale laxissima*.

**Figure 17 molecules-28-02503-f017:**
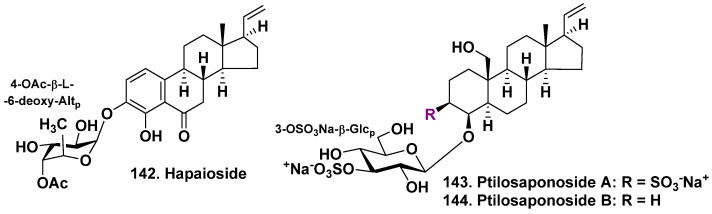
Hapaioside **142** from *Cribrochalina olemda* and ptilosaponosides **143**, **144** from *Ptilocaulis spiculifer*.

**Table 1 molecules-28-02503-t001:** Taxonomical distribution and biological activities of tetracyclic triterpene glycosides in sponges of the class Demospongiae.

Taxon	Place of Collection	Glycosides	Type of Biological Activity ^a^	Reference
Order Tetractinellida (Suborder Astrophorina)				
Family Geodiidae				
*Melophlus* sp.	Guam Island, Truk Lagoon, Micronesia	Sarasinoside A_1_ (**1**)	Cytotoxicity on lymphocytic leukemia P388 cell line (**1**)	Schmitz et al. [22]
*Melophlus sarasinorum*	Palau Islands, Micronesia	Sarasinosides A_1_ (**1**), B_1_ (**4**), C_1_ (**7**)	Ichthyotoxicity against killifish *P. reticulata*, inhibitory on fertilized eggs of the starfish *A. pectinifera* (**1**, **4**)	Kitagawa et al. [21]
-	-	Sarasinosides A_1_–A_3_ (**1**–**3**), B_1_–B_3_ (**4**–**6**), C_1_–C_3_ (**7**–**9**)	Ichthyotoxicity against *P. reticulata*, inhibitory on cell division of fertilized eggs of the starfish *A. pectinifera* (**1**, **4**)	Kobayashi et al. [24]
-	Solomon Islands	Sarasinosides B_1_ (**4**), D–G (**10**–**13**)	-	Espada et al. [25]
-	Guam Island, Micronesia	Sarasinosides A_1_ (**1**), A_3_ (**3**), H_1_ (**14**), H_2_ (**15**), I_1_ (**16**), I_2_ (**17**)	Cytotoxicity on human leukemia K562 cell line (**2**, **3**)	Lee et al. [26]
-	Sulawesi Island, Indonesia	Sarasinosides A_1_ (**1**), A_3_ (**3**), H_2_ (**15**), I_1_ (**16**), I_2_ (**17**), J–M (**18**–**21**)	Antimicrobial toward *B. subtilis* and *S. cerevisiae* (**1**, **18**)	Dai et al. [23]
-	Reef Scott, north-western coast of Australia	Sarasinosides A_1_–A_3_ (**1**–**3**), L (**20**), M (**21**), A_4_ (**22**), A_5_ (**23**)	-	Santalova et al. [27]
Unidentified sponge	Solomon Islands	Sarasinosides B_1_ (**4**), M_2_ (**24**)	Cytotoxicity toward Neuro-2a and HepG2 tumor cell lines (**24**)	Puilingi et al. [28]
Family Geodiidae				
*Erylus lendenfeldi*	Gulf of Eilat, Red Sea	Eryloside A (**25**)	-	Carmely et al. [29]
-	Gulf of Aqaba, Red Sea	Erylosides A (**25**), K (**26**), L (**27**)	Antibacterial against *B. subtilis* and *E. coli* (**25**); antifungal against *C. albicans* (**25**); brine shrimp assay (**25**, **26**)	Fouad et al. [30]
-	North of Hurghada, Red Sea	Erylosides A (**25**), K (**26**), L_1_ (**28**)	Cytotoxicity against a yeast strain (Δrad50) (**25**, **26**, **28**)	Sandler et al. [31]
*Erylus* sp.	South of New Caledonia	Erylosides C (**29**), D (**30**)	-	D’Auria et al. [32]
*Erylus goffrilleri*	Port Nelson, Rum Cay, Bahama Islands	Eryloside E (**31**)	Immunosuppressive in the mixed lymphocyte reaction assay (**31**)	Gulavita et al. [33]
-	Arresife-Seko Reef, Cuba	Erylosides R–U (**32**–**35**), F_5_–F_7_ (**36**–**38**), V (**39**)	Cytotoxicity against tumor Ehrlich carcinoma cells (**32**–**34**, **37**–**39**)	Afiyatullov et al. [34]
-	-	Erylosides F_8_ (**40**), V_1_–V_3_ (**41**–**43**), W (**44**), W_1_ (**45**), W_2_ (**46**)	Cytotoxicity against ascite form of Ehrlich carcinoma tumor cells, hemolysis (**40**–**46**)	Antonov et al. [35]
*Erylus formosus*	Bahamas Islands	Formoside (**47**)	-	Jaspars et al. [36]
-	-	Formoside (**47**), formoside B (**48**)	Antiviral against HSV-1, antibacterial against *C. xerosis*, antifungal against amphotericin B-resistant *C. abicans* (**47**)	Kubanek et al. [37]
-	-	Eryloside F (**49**)	Receptor antagonist activity, inhibition of platelet aggregation (**49**)	Stead et al. [38]
-	Puerto Morelos, Caribbean Sea, Mexico	Erylosides F (**49**), F_1_–F_4_ (**50**–**53**), M–Q (**54**–**58**), H (**67**)	Induction the early apoptosis of Ehrlich carcinoma cells (**52**); activation of the Ca^2+^ influx into mouse spleenocytes (**49**, **50**)	Antonov et al. [39]
-	-	Formoside (**47**), erylosides R_1_ (**59**), T_1_–T_6_ (**60**–**65**)	-	Antonov et al. [40]
*Erylus nobilis*	Jaeju Island, Republic of Korea	Erylosides G–J (**66**–**69**)	Cytotoxicity on the human leukemia cell line K562 (**66**–**69**)	Shin et al. [41]
-	Shikine-jima Island, 200 km south of Tokyo, Japan	Nobiloside (**70**)	Inhibition of neuraminidase from the bacterium *C. perfringens* (**70**)	Takada et al. [42]
*Erylus placenta*	Hachijo Island, South Japan	Sokodosides A (**71**), B (**72**)	Growth-inhibition against the fungus *M. ramanniana* and the yeast S. *cerevisiae* with and without mutations, cytotoxic against lymphocytic leukemia P388 cells (**71**, **72**)	Okada et al. [43]
Order Bubarida				
Family Dictyonellidae				
*Lipastrotethya* sp.	Chuuk Lagoon (Truk Lagoon), Micronesia	Sarasinosides A_1_ (**1**), A_2_ (**2**), H_1_ (**14**), H_2_ (**15**), J (**18**), M (**21**), N−R (**73**−**77**)	Cytotoxicity against A549 (**1**, **2**, **14**, **18**, **21**, **73**−**77**) and K562 (**1**, **2**, **14**, **15**, **18**, **21**, **73**−**76**) tumor cell lines; inhibition of Na^+^/K^+^-ATPase (**1**, **73**)	Lee et al. [44]
-	-	Sarasinosides A_1_ (**1**) A_3_ (**3**), B_2_ (**5**), M (**21**), A_4_ (**22**), Q (**76**), R (**77**), H_2_ (**15**), I_1_ (**16**), I_2_ (**17**), “glycoside 1” (**78**)	Cytotoxicity against ACHN, MDA-MB-231, NCI-H23, and NUGC-3 tumor cell lines (**1**, **3**, **5**, **22**, **76**, **78**)	Eom et al. [45]
*Dictyonella marsilii*	Coasts of Ceuta, Gibraltar Strait	Eryloside W (**79**)	-	Genta-Jouve et al. [46]
Order Haplosclerida				
Family Petrosiidae				
*Petrosia* sp.	North Sulawesi, Indonesia	Sarasinosides A_1_ (**1**), I_1_ (**16**), J (**18**), S (**80**)	-	Maarisit et al. [47]
*Petrosia nigricans*	Lipata, Surigao City, Philippines	5,8-Epoxysarasinoside (**81**), 8,9-epoxysarasinoside (**82**), sarasinosides A_1_ (**1**), H_1_ (**14**), I_1_ (**16**), I_2_ (**17**), O–R (**74**–**77**)	Cytotoxicity against HCT116 and A549 cancer cell lines (**81**, **82**)	Mama et al. [48]
Order Poecilosclerida				
Family Esperiopsidae				
*Ulosa* sp.	North-western coast of Madagascar Island	Ulososide A (**83**)	-	Antonov et al. [49]
-	-	Ulososide B (**84**)	-	Antonov et al. [50]
-	-	Ulososides C–E (**85**–**87**)	-	Antonov et al. [51]
Family Raspailiidae				
*Ectyoplasia ferox*	San Salvador Island, Bahamas	Ectyoplasides A (**88**), B (**89**)	Cytotoxicity against J774, WEHI164, and P388 murine cell lines (**88**, **89**)	Cafieri et al. [52]
-	Grand Bahama Island	Feroxosides A (**90**), B (**91**)	Cytotoxicity against J774 murine cell line (**90**, **91**)	Campagnuo-lo et al. [53]
-	Uraba Gulf, Colombia	Ulososides A (**83**), F (**92**), urabosides A (**93**), B (**94**)	-	Colorado et al. [54]

^a^ Using cell lines: P388—murine lymphocytic leukemia cell line; K562—human leukemic cell line; Neuro 2a—mouse neuroblastoma cell line; HepG2—human liver cancer cell line; A549—human lung adenocarcinoma cell line; ACHN—human renal adenocarcinoma; MDA-MB-231—human triple-negative breast cancer cell line; NCI-H23—human non-small lung carcinoma cell line; NUGC-3—human stomach adenocarcinoma cell line; HCT116—human colon cancer cell line; J774—murine monocyte/macrophage cell line; WEHI164—murine fibrosarcoma cell line.

**Table 2 molecules-28-02503-t002:** Taxonomical distribution and biological activities of steroid glycosides in sponges of the class Demospongiae.

Taxon	Place of Collection	Glycosides	Type of Biological Activity ^a^	Reference
Order Tetractinellida (suborder Astrophorina)				
Family Pachastrellidae				
*Pachastrella* sp.	Kamagi Bay, Ehime Prefecture, Japan	Pachastrelloside A (**95**)	Inhibition of cell division of fertilized starfish *A. pectinifera* eggs (**95**)	Hirota et al. [63]
*Pachastrella scrobiculosa*	Miura Peninsula, Japan	Scrobiculosides A (**96**), B (**97**)	Cytotoxicity against the mouse lymphoma cell line P388 and the human lymphoma cell line HL-60 (**96**, **97**)	Jomori et al. [64]
Families Vulcanellidae and Ancorinidae				
*Poecillastra wondoensis* and *Rhabdastrella (=Jaspis) wondoensis*	Cheju Island, Korea	Wondosterols A–C (**98**–**100**)	Cytotoxicity against P388 murine leukemia cells (**98**–**100**); antibacterial against *P. aeruginosa* and *E. coli* (**98**, **100**)	Ryu et al. [65]
*Poecillastra compressa*	Mediterranean Sea, French coast	Poecillastrosides A–G (**101**–**107**)	Antifungal against *A. fumigatus* (**104**, **105**)	Calabro et al. [66]
Order Poecilosclerida				
Family Microcionidae				
*Pandaros acanthifolium*	Canyon rock, Martinique Island, Caribbean Sea	Pandarosides A–D (**108**–**111**), methyl esters of pandarosides A (**108a**), C (**110a**), D (**111a**)	-	Cachet et al. [67]
-	-	Pandarosides E–J (**112**–**117**), methyl esters of pandarosides E–J (**112a**–**117a**)	Antiprotozoal against *T. b. rhodesiense*, *T. cruzi*, *L. donovani*, and *P. falciparum*; cytotoxicity against L6 cells (**112**–**117**, **112a**–**117a**, **108**–**111**, **108a**, **110a**, **111a**)	Regalado et al. [68]
-	-	Pandarosides K–M (**118**–**120**), methyl esters of pandarosides K–M (**118a**–**120a**)	Antiprotozoal against *T. b. rhodesiense*, *T. cruzi*, *L. donovani*, and *P. falciparum*; cytotoxicity against L6 cells, lung carcinoma NSCLC A549, colon carcinoma HT29, and breast MDA-MB-231 cells (**118**–**120**, **118a**–**120a**, **114**, **114a**); haemolytic (**121**, **122**, **124**)	Regalado et al. [69]
-	-	Acanthifoliosides A–F (**121**–**126**), methyl ester of acanthifolioside F (**126a**)	Antiprotozoal against *T. b. rhodesiense*, *T. cruzi*, *L. donovani*, and *P. falciparum*; cytotoxicity against L6 cells (**121**–**126**, **126a**)	Regalado et al. [70]
-	Florida Keys (USA, Florida)	Acanthifoliosides G–J (**127**–**130**)	Antioxidant and cytoprotective (**127**)	Berrué et al. [71]
Family Mycalidae				
*Mycale laxissima*	San-Felipe Island, Cuba	Mycaloside A (**131**)	-	Kalinovsky et al. [73]
-	-	Mycalosides B–I (**132**–**139**)	Inhibition the fertilization of eggs by sperm of the sea urchin *S. nudus* (**131**, **137**)	Antonov et al. [74]
-	-	Mycalosides J (**140**), K (**141**)	-	Afiyatullov et al. [75]
Order Haplosclerida				
Family Niphatidae				
*Niphates (=Cribrochalina) olemda*	Pohnpei, Micronesia	Hapaioside (**142**)	-	Yeung et al. [76]
Order *Axinellida*				
Family Axinellidae				
*Ptilocaulis spiculifer*	New Georgia Island (North East Kolingo), Solomon Islands	Ptilosaponosides A (**143**), B (**144**)	-	Gabant et al. [77]

^a^ Using cell lines: P388—murine lymphocytic leukemia cell line; HL-60—human leukemia cell line; L6—rat myoblast cell line; NSCLC A549—lung carcinoma cell line; HT29—colon carcinoma cell line; MDA-MB-231—human triple-negative breast cancer cell line.

## Data Availability

Data is contained within the article.

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
