# Peer review of "Triterpene and Steroid Glycosides from Marine Sponges (Porifera, Demospongiae): Structures, Taxonomical Distribution, Biological Activities"

_molecules, 2023, doi:10.3390/molecules28062503_

Round 1

Reviewer 1 Report

Dear Editor,

I would like to thank you for inviting me to review the paper. The study is interesting but some explanations for the results are missing, as well as the significance of this research in relation to some previous ones. Here enclose you can find my comments. If the authors accept the suggestions and make some corrections, then I suggest that this paper can be accepted for publication in Molecules.

1. The abstract should be rearranged. A good number of sentences should be transferred to the Introduction section. The abstract should be rearranged so that it corresponds to the goals and results of the review paper presented by the authors.

2. The Introduction is very poorly written. In this part, the importance of the topic and molecules as the species under investigation should be emphasized. A few more important references should be added that would add importance to this review.

3. The authors took into consideration an enviable number of previously published results. The review paper is written correctly, however, presenting a huge number of results without any logical order can be very confusing for readers. The proposal would be to group the results into several sub-headings taking into account the investigated biological activities of the isolated compounds.

4. The question arises whether there are any significant results obtained in In Vivo tests?

5. In the main part of the review work, there is not much discussion and some logical explanation about the obtained biological tests, the proposal is to add in the Conclusions part the importance of some of the most prominent results that were previously obtained, as well as their importance for some future tests.

6. A list of abbreviations would significantly improve readers' understanding of certain concepts. I put special emphasis on the cell lines that were used to test the antitumor potential.

Author Response

Dear Editor,

I would like to thank you for inviting me to review the paper. The study is interesting but some explanations for the results are missing, as well as the significance of this research in relation to some previous ones. Here enclose you can find my comments. If the authors accept the suggestions and make some corrections, then I suggest that this paper can be accepted for publication in Molecules.

Note 1: The abstract should be rearranged. A good number of sentences should be transferred to the Introduction section. The abstract should be rearranged so that it corresponds to the goals and results of the review paper presented by the authors.

Reply: We have revised the abstract in accordance with the comment of the Reviewer

Note 2: The Introduction is very poorly written. In this part, the importance of the topic and molecules as the species under investigation should be emphasized. A few more important references should be added that would add importance to this review.

Reply: We have revised the Introduction in accordance with the comment of the Reviewer and added references increasing the relevance of the review.

Note 3: The authors took into consideration an enviable number of previously published results. The review paper is written correctly, however, presenting a huge number of results without any logical order can be very confusing for readers. The proposal would be to group the results into several sub-headings taking into account the investigated biological activities of the isolated compounds.

Reply: We have grouped all the results according to the taxonomic position of the studied sponge species. The grouping of the described structures according to the biological activities they exhibit is not possible, since different types of biological activity are shown for the same compounds (see Tables 1 and 2, Type of Biological Activity)

Note 4: The question arises whether there are any significant results obtained in In Vivo tests?

Reply: These are not our own experiments. We only have provided all the data presented by the authors of original publications. The in vivo tests on the fish, for example, seems to be significant. The more detail information should be found in the original publications.

Note 5: In the main part of the review work, there is not much discussion and some logical explanation about the obtained biological tests, the proposal is to add in the Conclusions part the importance of some of the most prominent results that were previously obtained, as well as their importance for some future tests.

Reply: The main task of the review is to present the structural diversity, taxonomic distribution and biological role of so interesting natural phenomenon as sponge saponins. All these points are adequately reflected in the Conclusions. Concerning the biological activities. There are no so prominent data and all the necessary information concerning the activities for each group of glycosides may be found easy by the readers in the table 1 and 2 and in corresponding parts of the article described the different glycosides. The titles of the Table 1 and 2 were expanded in order to simplify the search of the data concerning biological activities.

Note 6: A list of abbreviations would significantly improve readers' understanding of certain concepts. I put special emphasis on the cell lines that were used to test the antitumor potential.

Reply: The cell lines are depicted after the table 1 and 2.

Reviewer 2 Report

  A moderate check is required for the English language and style through the text, for instance:  Abstract: The article is a comprehensive review concerning tetracyclic triterpene and steroid glycosides from sponges (Porifera, Demospongia). The findings of the saponins outside of higher plants are very rare and only several examples are occurred in the nature including steroidal glycosides from starfish, sea cucumber triterpene glycosides, and steroid and terpenoid glycosides from the sponges. The high structural diversity of the sponge saponins is the result '

2.      The list of keywords could be expanded to capture the main findings.

3.      It needs to improve the objectives and rationale of the study.

4.      Line 31: it is worth clarifying 'Terpenoid and steroid glycosides in marine sponge were discovered later' (When?)

5.      In the introduction, it is worth citing modern data on various groups of bioactive components that can accumulate in marine sponges. To do this, it is worth analyzing articles from the last 5 years, which are easy to find in the scientometric databases Scopus/PubMed, for example:

·  Galitz, A., Nakao, Y., Schupp, P. J., Wörheide, G., & Erpenbeck, D. (2021). A Soft Spot for Chemistry-Current Taxonomic and Evolutionary Implications of Sponge Secondary Metabolite Distribution. Marine drugs19(8), 448. https://doi.org/10.3390/md19080448

·  Nabil-Adam, A., Shreadah, M. A., El Moneam, N. M. A., & El-Assar, S. A. (2020). Various In Vitro Bioactivities of Secondary Metabolites Isolated from the Sponge Hyrtios aff. Erectus from the Red Sea Coast of Egypt. Turkish journal of pharmaceutical sciences17(2), 127–135. https://doi.org/10.4274/tjps.galenos.2018.72677

6.      Generally, the review should be supplemented so that there are at least 100 processed sources in total. Such additions would significantly strengthen the novelty and rationale of the chosen topic.

7.      Table 1 is mentioned firstly on page 2, but it is given as far as page 23. This is worth fixing.

8.      There is the Section  '5. Biological Roles of Sponge Triterpene Glycosides',

9.      The list of sources does not list any scientific publications for 2022, although the Conclusions (line 792) give the following generalization: 'About 142 sponge saponins were isolated by the end of 2022, including' .... It should be corrected.

TThe Conclusions section should be shortened.

1Line 995, etc. - The italic type should be used everywhere for writing Latin names of genera and species, for instance:  Erylus formosus

Author Response

Note 1: A moderate check is required for the English language and style through the text, for instance: Abstract: The article is a comprehensive review concerning tetracyclic triterpene and steroid glycosides from sponges (Porifera, Demospongia). The findings of the saponins outside of higher plants are very rare and only several examples are occurred in the nature including steroidal glycosides from starfish, sea cucumber triterpene glycosides, and steroid and terpenoid glycosides from the sponges. The high structural diversity of the sponge saponins is the result '

Reply: The English was carefully checked including the Abstract that was significantly compressed because of strict recommendation of another referee.

Note 2: The list of keywords could be expanded to capture the main findings.

Reply: The list of the keywords is expanded

Note 3: It needs to improve the objectives and rationale of the study.

Reply: The Introduction was significantly expanded in order to explain the objectives of the review.

Note 4: Line 31: it is worth clarifying 'Terpenoid and steroid glycosides in marine sponge were discovered later' (When?)

Reply: The glycosides were discovered in 1980-ies–beginning of 1990-ies. The corresponding phrase is added. The more detail information is provided in the main text.

Note 5: In the introduction, it is worth citing modern data on various groups of bioactive components that can accumulate in marine sponges. To do this, it is worth analyzing articles from the last 5 years, which are easy to find in the scientometric databases Scopus/PubMed, for example:

Galitz, A., Nakao, Y., Schupp, P. J., Wörheide, G., & Erpenbeck, D. (2021). A Soft Spot for Chemistry-Current Taxonomic and Evolutionary Implications of Sponge Secondary Metabolite Distribution. Marine drugs, 19(8), 448. https://doi.org/10.3390/md19080448

Nabil-Adam, A., Shreadah, M. A., El Moneam, N. M. A., & El-Assar, S. A. (2020). Various In Vitro Bioactivities of Secondary Metabolites Isolated from the Sponge Hyrtios aff. Erectus from the Red Sea Coast of Egypt. Turkish journal of pharmaceutical sciences, 17(2), 127–135. https://doi.org/10.4274/tjps.galenos.2018.72677

Reply: We have expanded the Introduction strictly along with this recommendation.

Note 6: Generally, the review should be supplemented so that there are at least 100 processed sources in total. Such additions would significantly strengthen the novelty and rationale of the chosen topic.

Reply: We have expanded the list of the references by 85 but its further expansion should be too artificial.

Note 7: Table 1 is mentioned firstly on page 2, but it is given as far as page 23. This is worth fixing.

Reply: We have expanded the reference in the text as “(see Section 4. Taxonomic Distribution of Glycosides in Sponges, Table 1)”.

Note 8: There is the Section '5. Biological Roles of Sponge Triterpene Glycosides',

Reply: The name of the section “5. Biological Roles of Sponge Triterpene Glycosides” is correct now.

Note: 9. The list of sources does not list any scientific publications for 2022, although the Conclusions (line 792) give the following generalization: 'About 142 sponge saponins were isolated by the end of 2022, including' .... It should be corrected.

Reply Many thanks for this comment. We already fixed it and added the data even for the beginning of 2023.

Note 10: The Conclusions section should be shortened.

Reply: We have carefully checked the Conclusion but did not find a possibility to short something. However, we have declined the requirement of another referee to expand the Conclusion.

Note 11: Line 995, etc. - The italic type should be used everywhere for writing Latin names of genera and species, for instance:  Erylus formosus

Reply: The error is fixed

Round 2

Reviewer 2 Report

There is the Section '5. Biological Roles of Sponge Triterpene Glycosides', but absent another necessary section '6. Biological Roles of Sponge  Steroid Glycosides'. It should be obviously added.

Author Response

Refreree 2, round 2

Note: There is the Section '5. Biological Roles of Sponge Triterpene Glycosides', but absent another necessary section '6. Biological Roles of Sponge Steroid Glycosides'. It should be obviously added.

Reply: Because biological role of sponge steroid glycoside was never specially studied there is no a sense to create a special section. Nevertheless, we have expanded the title of this section and added the additional paragraph concerning steroid glycosides. All the changes are marked with yellow. Many thanks for this note.